# GWAS and functional studies suggest a role for altered DNA repair in the evolution of drug resistance in *Mycobacterium tuberculosis*

**Saba Naz[1,2,3†], Kumar Paritosh[4†], Priyadarshini Sanyal[1], Sidra Khan[1], Yogendra Singh[3], Umesh Varshney[5], Vinay Kumar Nandicoori[1,2]***

[1]National Institute of Immunology, New Delhi, India; [2]Centre for Cellular and Molecular Biology, Hyderabad, India; [3]Department of Zoology, University of Delhi, Delhi, India; [4]Centre for Genetic Manipulation of Crop Plants, University of Delhi South Campus, New Delhi, India; [5]Department of Microbiology and Cell Biology, Indian Institute of Science Bangalore, Bangalore, India

**\*For correspondence:**
vinaykn@nii.ac.in

[†]These authors contributed equally to this work

**Competing interest:** The authors declare that no competing interests exist.

**Abstract** The emergence of drug resistance in *Mycobacterium tuberculosis* (*Mtb*) is alarming and demands in-depth knowledge for timely diagnosis. We performed genome-wide association analysis using 2237 clinical strains of *Mtb* to identify novel genetic factors that evoke drug resistance. In addition to the known direct targets, we identified for the first time, a strong association between mutations in DNA repair genes and the multidrug-resistant phenotype. To evaluate the impact of variants identified in the clinical samples in the evolution of drug resistance, we utilized knockouts and complemented strains in *Mycobacterium smegmatis* and *Mtb*. Results show that variant mutations compromised the functions of MutY and UvrB. MutY variant showed enhanced survival compared with wild-type (*Rv*) when the *Mtb* strains were subjected to multiple rounds of ex vivo antibiotic stress. In an in vivo guinea pig infection model, the MutY variant outcompeted the wild-type strain. We show that novel variant mutations in the DNA repair genes collectively compromise their functions and contribute to better survival under antibiotic/host stress conditions.

## Editor's evaluation

This paper provides important evidence implicating polymorphisms in the mycobacterial adenine DNA glycosylase, MutY, in the emergence of antibiotic resistance in *Mycobacterium tuberculosis*. While the precise mechanism underlying this phenotype requires further investigation, the inference from genome-wide association analyses of sequenced clinical isolates, supported by laboratory experiments and animal infection models, is convincing. This work adds a new locus of interest to the list of polymorphisms associated with tuberculosis drug resistance, and is likely to be relevant to the mycobacterial research field.

## Introduction

The acquisition of drug resistance in *Mycobacterium tuberculosis* (*Mtb*) has evoked a perilous situation worldwide (*WHO, 2020*). Resistance to isoniazid and rifampicin, the first-line drugs, results in **m**ulti**d**rug **r**esistant-TB (MDR-TB). The pathogen is defined as e**x**tensively **d**rug **r**esistant (XDR) when it becomes resistant to first-line TB drugs, any fluoroquinolones, and at least one additional Group A drug (moxifloxacin, levofloxacin, linezolid, and bedaquiline) (*WHO, 2021*). Prolonged treatment

duration, high drug toxicity, and the expensive drug regimen pose a challenge for treating the MDR and XDR-TB. Moreover, the inadequate treatment of drug-resistant TB leads to the augmentation of resistance to other anti-TB drugs, increasing the probability of transmission of these strains in the population (*Alexander and De, 2007*; *Bastos et al., 2014*).

Seven major lineages of *Mtb* are present across the globe, out of which four lineages: Lineage 1-Indo Oceanic (EAI); Lineage 2- Beijing; Lineage 3- Central Asian (CAS); and Lineage 4- Euro-American (*Gagneux et al., 2006*) are prevalent in humans. Clinical strains belonging to lineage 2 are more prone to developing drug resistance than lineage 4 strains (*Ford et al., 2013*). Acquisition of drug resistance in *Mtb* is majorly attributed to the chromosomal mutations that either modify the antibiotic's direct target or increase the expression of efflux pumps that helps in decreasing the effective concentration of the drug inside the cell. The expression of drug modifying/degrading enzymes also contributes to the acquisition of drug resistance (*Gygli et al., 2017*). Despite well-known mechanisms of drug resistance, it is difficult to predict the resistance based on direct target mutations alone, implying the presence of hitherto unknown mechanisms that impart drug resistance. In addition, the diagnosis based on mutations in a particular known target region increases the bias that may have multiple repercussions, such as misdiagnosis and eventual spread of drug resistance (*CRyPTIC Consortium and the 100,000 Genomes Project et al., 2018*; *The CRyPTIC Consortium, 2022b*; *The CRyPTIC Consortium, 2022a*). The current knowledge of the mechanisms and biological triggers involved in the evolution of MDR or XDR-TB is inadequate. This knowledge is crucial for developing new drug targets and improved diagnosis. Multiple efforts have been made to determine the mechanisms for the emergence of MDR and XDR-TB. Genome-wide association studies (GWAS) identified different genes that abet the emergence of drug resistance (*Farhat et al., 2013*; *Hicks et al., 2018*; *Zhang et al., 2013*; *Safi et al., 2019*). However, only a few genes, such as *ponA1, prpR, ald, glpK,* and the mutation in the *thyA-Rv2765 thyX-hsdS.1* loci, are validated (*Farhat et al., 2013*; *Hicks et al., 2018*; *Zhang et al., 2013*; *Safi et al., 2019*).

In a quest to identify genetic triggers that aid in the evolution of antibiotic resistance in *Mtb*, we performed GWAS using global data set of 2237 clinical strains that consist of antibiotic susceptible, MDR, poly-drug resistant (resistant to more than one first-line anti-TB drug other than both isoniazid and rifampicin), pre-XDR, and XDR. Interestingly, we have identified mutations in the multiple DNA repair genes of *Mtb* associated with the MDR phenotype. Functional validation of the identified mutations in DNA repair enzymes revealed that perturbations in the DNA repair mechanisms result in the enhanced survival of strains in the presence of antibiotics ex vivo and in vivo.

## Results
### GWAS unveils mutations in the DNA repair genes

To identify the genetic determinants contributing to the development of antibiotic resistance in *Mtb*, we performed genome-wide association analysis using the whole-genome sequences of clinical strains from nine published studies. After the quality filtering of raw reads, the dataset had 2773 clinical strains from 9 different countries belonging to all 4 lineages (*Figure 1a*, *Figure 1—figure supplement 1*; *Hicks et al., 2018*; *Zhang et al., 2013*; *Casali et al., 2014*; *Blouin et al., 2012*; *Shanmugam et al., 2019*; *Guerra-Assunção et al., 2015*; *Clark et al., 2013*; *Bryant et al., 2013*; *Walker et al., 2013*). The dataset chiefly represented Lineage 2 and 4 isolates that are predominant across the globe (*Figure 1b*). Based on the computational predictions and the phenotypes provided by the previous studies, strains were categorized as susceptible, mono-drug resistant, MDR, Poly-DR, and pre-XDR (*Supplementary file 1*; *Supplementary file 2Manson et al., 2017*). We identified ~160,000 **s**ingle **n**ucleotide **p**olymorphisms (SNPs) and indels after mapping the short reads on the reference *Rv* genome. A phylogenetic tree constructed using the SNPs shows the proper clustering of lineages (*Figure 1b*). The total number of SNPs observed for all the strains was comparable, suggesting the absence of genetic drift (*Figure 1c*).

We hypothesized that the probability of finding the genetic determinants contributing to drug resistance would be higher in the strains resistant to more than two antibiotics. Thus, we performed GWAS using 1815 drug-susceptible and 422 drug-resistant strains (*Figure 1—figure supplement 1*, *Figure 2—figure supplements 1–4* & *Supplementary file 2*). We employed a **g**enome **a**ssociation and **p**rediction **i**ntegrated **t**ool (GAPIT) software, with stringent false discovery rate (FDR) adjusted p-value

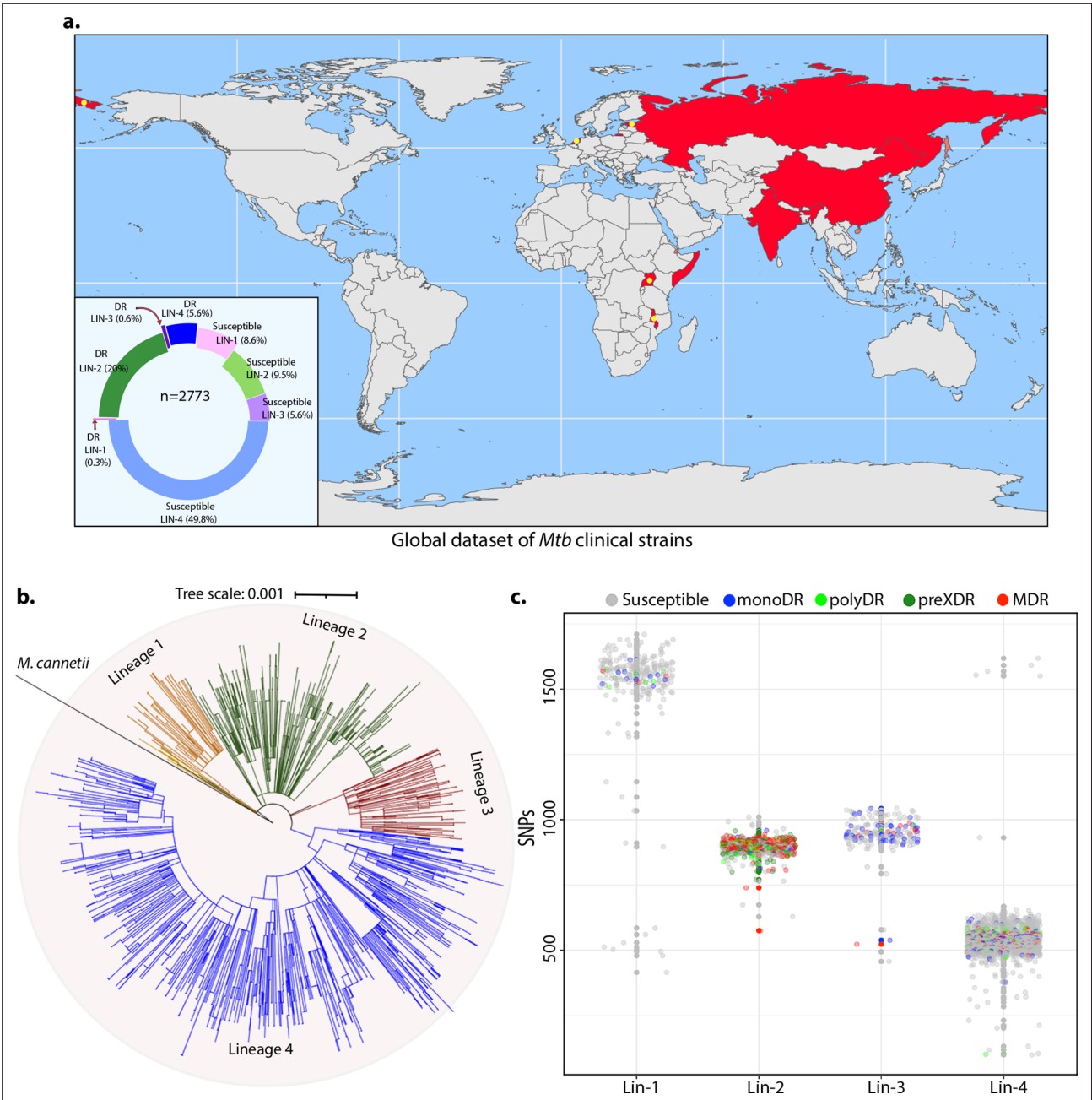

**Figure 1.** Genome-wide association study unveils mutations in the DNA repair genes. (**a**) Geographical distribution of 2773 clinical strains of *Mycobacterium tuberculosis* (*Mtb*). The donut plot represents the proportion of susceptible and drug-resistant (DR) strains in each lineage. DR includes mono-DR, poly-DR, multidrug resistant (MDR), and pre-extensively drug resistant (XDR). A detailed breakup of distribution is given in **Supplementary file 1**. (**b**) Phylogenetic tree constructed using 1,60,000 single nucleotide polymorphisms (SNPs) using *Mycobacterium canetti* as an outgroup. (**c**) Dot-plot showing the number of SNPs identified in each strain. Different colored dots indicate the drug resistance phenotype of strain.

The online version of this article includes the following figure supplement(s) for figure 1:

**Figure supplement 1.** Country-wide distribution of clinical strains.

(***Gao et al., 2016***; ***Zegeye et al., 2014***). After setting the adjusted p-value cut-off at $10^{-5}$, we identified 188 mutations, including 24 intergenic regions correlated with multidrug resistance (***Supplementary file 3***; ***Supplementary file 4***; ***Supplementary file 5***; ***Supplementary file 6***). The effect of identified SNPs on the development of MDR/XDR reveals positive or negative contributions (***Figure 2a***). We have identified known first- and second-line drug resistance target genes (***Figure 2b*** & ***Table 1***).

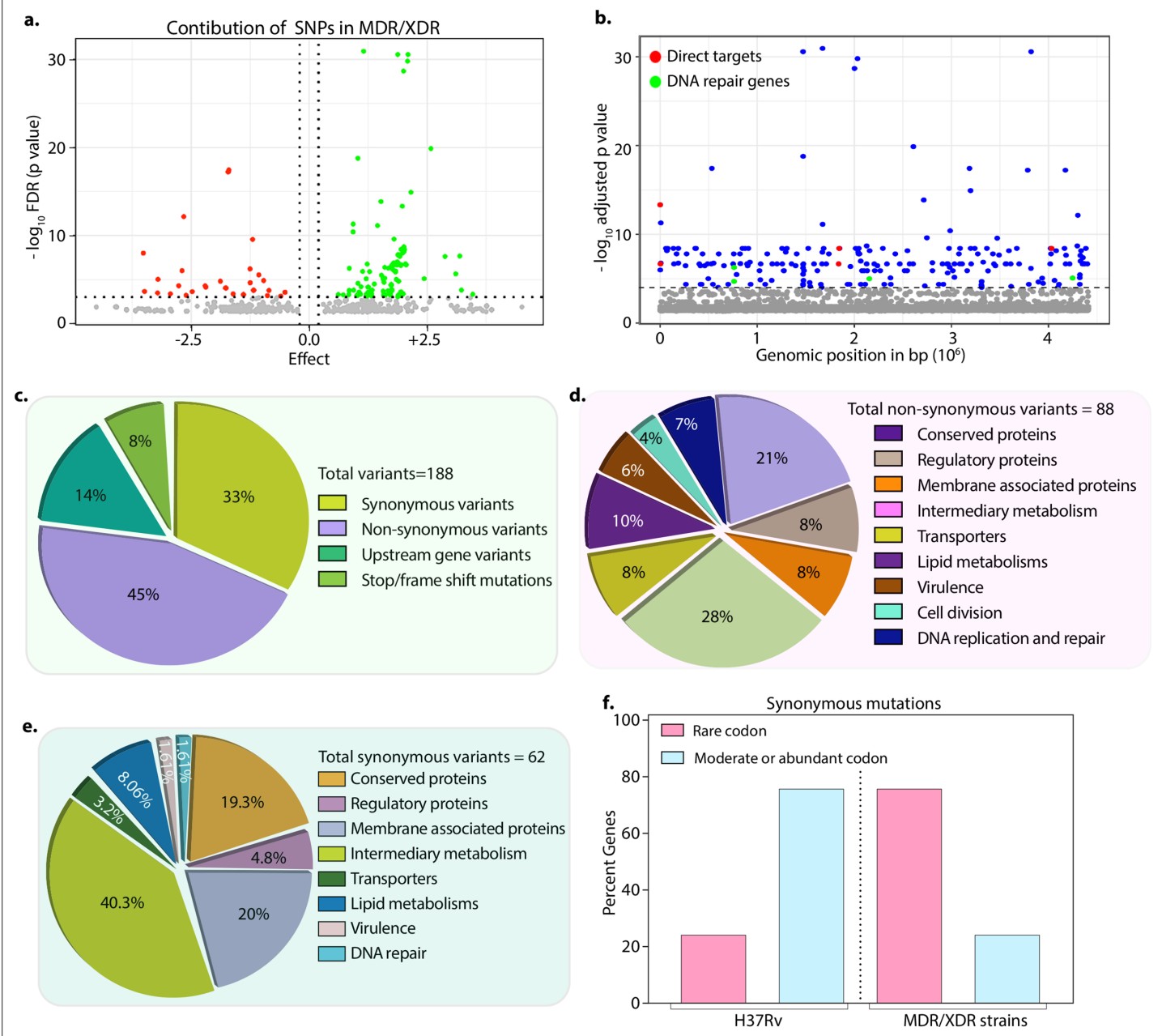

**Figure 2.** Drug-resistant strains carry mutations in the DNA repair genes. (**a**) Volcano plot represents the effect of identified single nucleotide polymorphisms (SNPs) on the development of multidrug resistant/extensively drug resistant TB (MDR/XDR-TB). The positive effect (green dots) shows that the identified SNPs would aid in MDR/XDR development. The negative effect (red dots) shows that the SNPs would restrain the development of MDR/XDR. (**b**) Manhattan plot representing the association between the genes and drug resistance phenotype. A total of 188 genes that include intergenic regions were identified above the $10^{-5}$ cut-off value through association studies. Blue dots represent mutation in the lipid metabolism, membrane proteins, intermediary metabolism genes, and others. Green dots represent mutation in the direct targets for the first- and second-line antibiotics. Red dots represent mutations associated with the DNA repair genes. A detailed list of associated genes is provided in *Supplementary file 3*; *Supplementary file 4*. (**c–e**) Pie chart represents the total (**c**), non-synonymous (**d**), and synonymous (**e**) SNPs identified in the genes that belong to different categories. (**f**) Bar plot represents the percentage of synonymous mutations in the genes that resulted in abundant/moderate codon usage to the rare codon compared to H37Rv.

The online version of this article includes the following source data and figure supplement(s) for figure 2:

**Source data 1.** Mutations identified in genes that belong to different categories.

**Figure supplement 1.** Genome-wide association study analysis.

**Figure supplement 2.** Linkage disequilibrium, minor allele frequency and quantile-quantile plot in the genome-wide association analysis.

*Figure 2 continued on next page*

Although we identified multiple mutations in *rpoB*, only p.Leu452Pro and p.Val496Met were above the cut-off. Notably, mutations in the *rrs, katG* (p.Ser315Thr), *embB* (p.Gly406Ser), *pncA* (p.His71Arg), *gyrA* (p.Ala90Val), and recently reported genetic determinants such as *folC* (p.Ser150Gly), and *pks* were part of the 164 genes, validating our approach (*Figure 2b* & *Table 1*, *Supplementary file 4*; *Supplementary file 6*). Also, we identified the known compensatory mutations in the *fabG1* upstream region, *eis-Rv2417c*, and *oxyR-ahpC* loci (*Coll et al., 2018*; *Supplementary file 6*). Importantly, the above mutations were absent in the mono-DR and drug-susceptible strains (*Figure 2—figure supplement 6*).

Among the 188 genes, 45% of the mutations resulted in non-synonymous changes, whereas 33% resulted in synonymous changes, 14% in the upstream regions of the genes, and 8% in the stop/frameshift mutations (*Figure 2c–e* & -- *Supplementary file 3*; *Supplementary file 4*; *Supplementary file 5*; *Supplementary file 6*). While the non-synonymous and the stop/frameshift mutations most likely affect the functions of the proteins, the intergenic region mutations may impact gene expression. We identified mutations in genes involved in lipid metabolism, intermediary metabolism and respiration, membrane transporters, cell wall and cell processes, membrane-associated proteins, and others (https://mycobrowser.epfl.ch/) (*Figure 2c–e*). Synonymous changes may alter the mRNA stability or stall the translation process by changing an abundant codon to a rare codon (*Brandis and Hughes, 2016*; *Kristofich et al., 2018*; *Plotkin and Kudla, 2011*). Analysis of the synonymous mutations for the codon bias revealed that in >50% of events, codons were converted from moderate/abundant to rare codons (compare 75% in MDR/XDR with 25% in H37Rv) (*Figure 2f, Supplementary file 7*).

In addition to the mutations described above, we identified novel mutations in base excision repair (BER), nucleotide excision repair (NER), and homologous recombination (HR) pathway genes, *mutY, uvrA, uvrB,* and *recF* that are associated with the MDR and XDR-TB (*Figure 2b* & *Table 2*). Mutations in the DNA repair pathway genes could contribute to the selection and evolution of antibiotic resistance (*Table 2*; *Figure 2—figure supplement 5*). Analysis showed that mutations in the DNA repair genes are distributed specifically in MDR, PDR, and preXDR/XDR strains (*Figure 2—figure supplement 6a–d*). Furthermore, these strains also harbored mutations in the direct targets of the antibiotics (*Figure 2—figure supplement 6i–k*). Collectively, in addition to mutations in the direct targets, we identified novel uncharacterized variants, including mutations in the DNA repair genes.

**Table 1.** Mutations identified in the direct targets of antibiotics.

| Antibiotic | Gene | Mutations identified |
|---|---|---|
| Rifampicin | *rpoB* | Leu452Pro, Val496Meth |
| Isoniazid | *katG* | Ser315Thr |
| Ethambutol | *embB* | Gly406Ser |
| Ofloxacin | *gyrA* | Ala90Val, Ser91Pro |
| Kanamycin | *rrs* | 7 independent mutations |
| Pyrazinamide | *pncA* | His71Arg |
| Ethionamide | *ethA* | Met95Arg, Pro160(frame-shift) |
| Streptomycin | *gidB* | Leu35 (frame-shift) |
| Cycloserine | *ald* | Thr427Pro |

## The mutations in DNA repair genes result in their deficient function

DNA repair pathways, including NER, HR, and BER, guard the genomic integrity (*Cole et al., 1998*; *Singh, 2017*). The *Mtb* MutY is a 302 amino

**Table 2.** Mutations in DNA repair genes associated with drug resistance phenotype.

| Gene | Amino acid change | Wild type | Mutated | False discovery rate-adjusted p-value |
|---|---|---|---|---|
| *mutY* | Arg262Gln | G | A | 3.83E-09 |
| *uvrB* | Ala524Val | C | T | 2.15E-07 |
| *uvrA* | Gln135Lys | C | A | 3.83E-09 |
| *RecF* | Gly269Gly | G | T | 2.15E-07 |

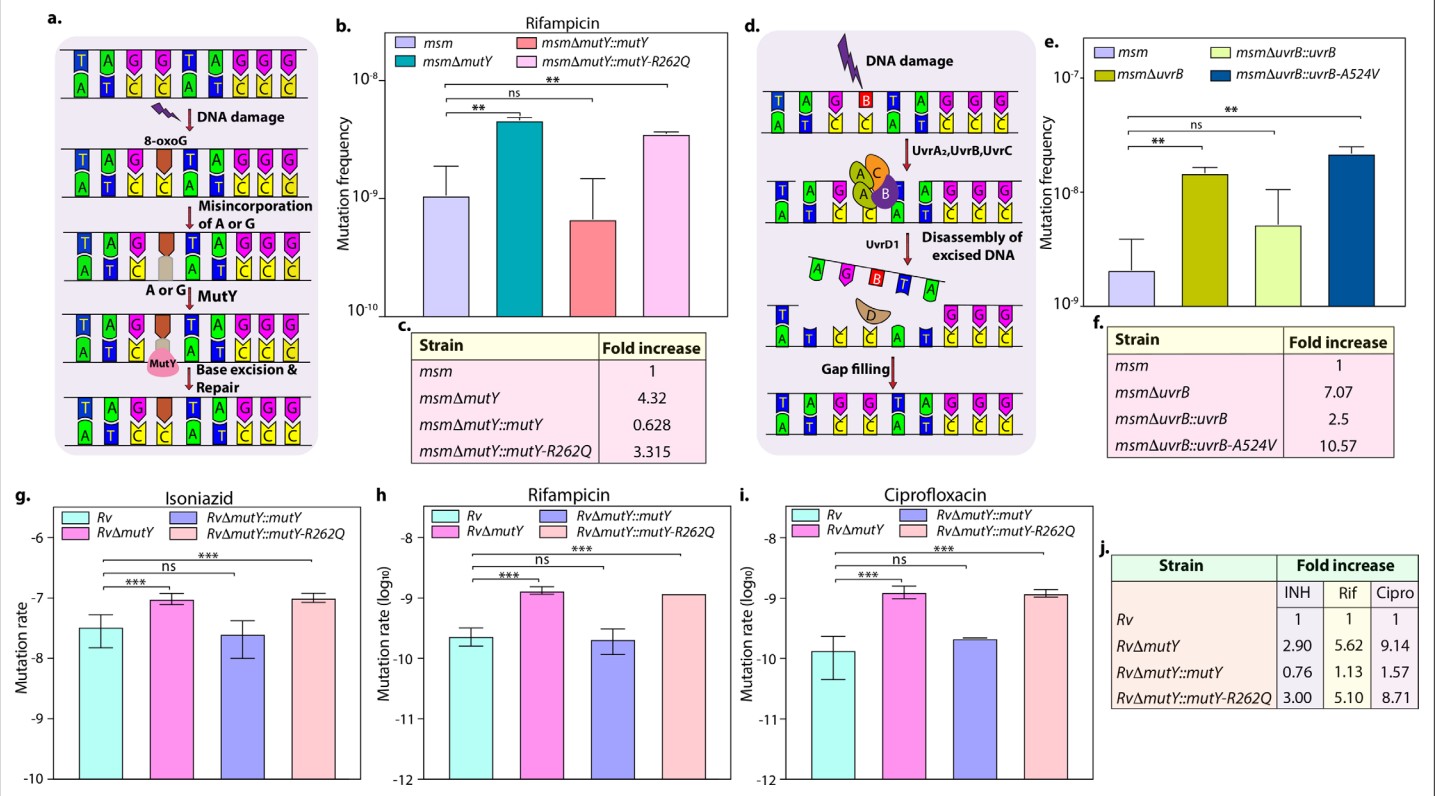

**Figure 3.** Variants identified in DNA repair genes abrogate their function. (**a**) A schematic representation of the base excision repair pathway that operates in mycobacteria. Oxidative damage can result in the conversion of G to 8-oxo-G. If MutM (Fpg) does not repair 8-oxo-G before replication, often an A is inserted against 8-oxo-G during replication. Under these conditions, MutM must avoid repair of 8-oxo-G until MutY removes the erroneously incorporated A. The predominant target of MutY is 8-oxo-G:A pair where it removes A; thus, the action of MutY provides another opportunity to incorporate C (the correct base) against 8-oxo-G. Now the DNA becomes a target for MutM again, leading to the removal of 8-oxo-G and allowing incorporation of G. (**b**) Mutation frequency was calculated using *msm, msmΔmutY, msmΔmutY::mutY, msmΔmutY::mutY-R262Q.* (**c**) Fold increase in the mutation frequency with respect to wild-type *msm*. (**d**) A schematic representation of the nucleotide excision repair pathway showing the recognition and initiation of repair by UvrA-UvrB and UvrC. (**e**) Mutation frequency of *msm, msmΔuvrB, msmΔuvrB::uvrB,* and *msmΔuvrB::uvrB-A524V.* (**f**) Fold increase in the mutation frequency with respect to wild-type *msm*. Two biologically independent experiment sets were performed. Each biological experiment was performed in a biological sextet. Data represent one set of experiments. Statistical analysis (two-way ANOVA) was performed using Graph pad prism software. ***$p<0.0001$, **$p<0.001$, and *$p<0.01$. (**g, h & i**) Mutation rate was calculated for different strains in the presence of isoniazid (**g**), rifampicin (**h**), or ciprofloxacin (**i**). (**j**) Table showing the fold increase in the mutation rate in comparison with wild-type *Rv*. The experiment was performed using six independent colonies. Data represent mean and standard deviation. Statistical analysis (two-way ANOVA) was performed using Graph pad prism software. ***$p<0.0001$, **$p<0.001$, and *$p<0.01$.

The online version of this article includes the following source data and figure supplement(s) for figure 3:

**Source data 1.** Mutation rate analysis in the presence of different drugs.

**Figure supplement 1.** Genome-wide association study of lineage 4 strains identified mutations in the DNA repair genes.

**Figure supplement 1—source data 1.** Confirmation of gene repalcement mutant and complementation strains.

**Figure supplement 2.** Mutation frequency analysis.

**Figure supplement 2—source data 1.** Analysis of Mutation frequency.

acid (aa) long adenine DNA glycosylase encoded by *rv3589*. We identified **Arg262Gln** mutation at the C-terminal region of the MutY. Oxidative damage to DNA results in the formation of 7,8-dihydro 8-oxoguanine (8-oxoG). If left unrepaired by MutM (fpg), it results in 8-oxoG:A (mostly) or 8-oxoG:G base pairing. MutY removes A or G paired against 8-oxoG, allowing MutM to correct the mistake. The absence of repair leads to G:C to T:A or C:G mutations in the genome (*Figure 3a*; *Kurthkoti et al., 2010*). The analysis of the mutation spectrum in the drug-resistant clinical strains harboring mutY-R262Q mutation and closely related drug-susceptible strains showed a bias toward C→A, A→G, and C→T mutations (*Figure 2—figure supplement 6I*). To decipher the biological role of the identified

variant, we cloned *Mtb mutY* and performed site-directed mutagenesis to generate the mutant allele. The wild type and mutant *mutY* genes were subcloned into an integrative *Mtb* shuttle vector. Constructs were electroporated into *Mycobacterium smegmatis mutY* mutant strain (*msmΔmutY*) to generate *msmΔmutY::mutY* and *msmΔmutY::mutY*-R262Q strains. We performed mutation frequency analysis to evaluate the impact of Arg262Gln mutation on its DNA repair function. In accordance with the published data, deletion resulted in a 4.32-fold increase in the mutation frequency (*Figure 3b*; *Kurthkoti et al., 2010*). While complementation with wild-type *mutY* rescued the phenotype, complementation with *mutY*-R262Q failed to do so (*Figure 3b and c*).

Next, we investigated the role of mutation in the NER pathway gene UvrB. UvrA, UvrB, and UvrC recognize and initiate the NER pathway upon DNA damage. UvrB, a 698 aa long DNA helicase encoded by *rv1633,* plays a pivotal role in the NER pathway by interacting with the UvrA and UvrC (*Figure 3d*; *Kurthkoti et al., 2008*). UvrB harbors N and C-terminal helicase domain, interaction domain, YAD/RRR motif, and UVR domain. Identified UvrB variant, **Ala524Val,** mapped to the C-terminal helicase like domain (*Theis et al., 2000*). To evaluate the functional significance of the mutation of *uvrB*, *Mtb uvrB* and *uvrB*-A524V genes were cloned into an integrative vector. The absence of *uvrB* led to higher mutation frequency, which could be rescued upon complementation with the wild type but not with the variant (*Figure 3e and f*).

Subsequently, we sought to extend our investigations in *Mtb*. Toward this, we generated the gene replacement mutant of *mutY* in laboratory strain *Mtb H37Rv* (*Rv*), wherein the *mutY* at native loci was disrupted with a hygromycin resistance cassette. Replacement at the native loci was confirmed by performing multiple PCRs (*Figure 3—figure supplement 1b–c*). Complementation constructs harboring *mutY* or *mutY*-R262Q were electroporated in the *RvΔmutY* to generate *RvΔmutY::mutY* and *RvΔmutY::mutY*-R262Q. Western blot analysis showed comparable expression of the MutY or MutY-R262Q (*Figure 3—figure supplement 1d*). We determined the mutation rates in the presence of isoniazid, rifampicin, and ciprofloxacin (*Figure 3g–j*). The fold increase in the mutation rates relative to *Rv* for *RvΔmutY*, *RvΔmutY:mutY*, and *RvΔmutY::mutY*-R262Q were 2.90, 0.76, and 3.0 in the presence of isoniazid; 5.62, 1.13, and 5.10 in the presence of rifampicin; and 9.14, 1.57, and 8.71 in the presence of ciprofloxacin, respectively (*Figure 3j*). Also, we have determined the mutation frequencies in the presence of isoniazid and rifampicin (*Figure 3—figure supplement 2*). Results are in line with the mutation rate experiments presented in *Figure 3*. Together these data suggest that variants of *mutY* and *uvrB* compromise their function.

## The variant of *mutY* resists antibiotic killing

The killing kinetics in the presence and absence of isoniazid, rifampicin, ciprofloxacin, and ethambutol was performed to evaluate the effect of different drugs on the survival of *RvΔmutY* or *RvΔmutY::mutY*-R262Q (*Figure 4a*). In the absence of antibiotics, the growth kinetics of *Rv*, *RvΔmutY*, *RvΔmutY::mutY*, and *RvΔmutY::mutY*-R262Q were similar (*Figure 4b*). In the presence of isoniazid, ~2 log-fold decreases in bacterial survival was observed on day 3 in *Rv* and *RvΔmutY::mutY*; however, in *RvΔmutY* and *RvΔmutY::mutY*-R262Q, the difference was limited to ~1.5 log-fold (*Figure 4c*). A similar trend was apparent on days 6 and 9, suggesting an ~fivefold increase in the survival of *RvΔmutY* and *RvΔmutY::mutY*-R262Q compared with *Rv* and *RvΔmutY::mutY* (*Figure 4c*). Interestingly, in the presence of ethambutol, we did not observe any significant difference (*Figure 4d*). In the presence of rifampicin and ciprofloxacin, we observed an ~10-fold increase in the survival of *RvΔmutY* and *RvΔmutY::mutY*-R262Q compared with *Rv* and *RvΔmutY::mutY* (*Figure 4e–f*). Thus, results suggest that the absence of *mutY* or the presence of *mutY* variant aids in subverting the antibiotic stress.

## The variant of *mutY* confers survival advantage ex vivo

Next, we evaluated the survival of *Rv*, *RvΔmutY*, *RvΔmutY::mutY,* and *RvΔmutY::mutY*-R262Q in the peritoneal macrophages. We did not observe any differences in the survival of *RvΔmutY* or *RvΔmutY::mutY*-R262Q compared with *Rv* or *RvΔmutY::mutY* (*Figure 5—figure supplement 1a–b*). We speculated that the evolution of a strain to become antibiotic-resistant requires the continued presence of antibiotic and host-directed stress. Therefore, we infected peritoneal macrophages with *Rv*, *RvΔmutY*, *RvΔmutY::mutY,* and *RvΔmutY::mutY*-R262Q in the absence or presence of the antibiotics. The bacteria recovered after 120 hr post-infection (p.i.) were cultured in vitro for 5 days and used for the next round of infection. The process was repeated for three rounds, and CFUs were

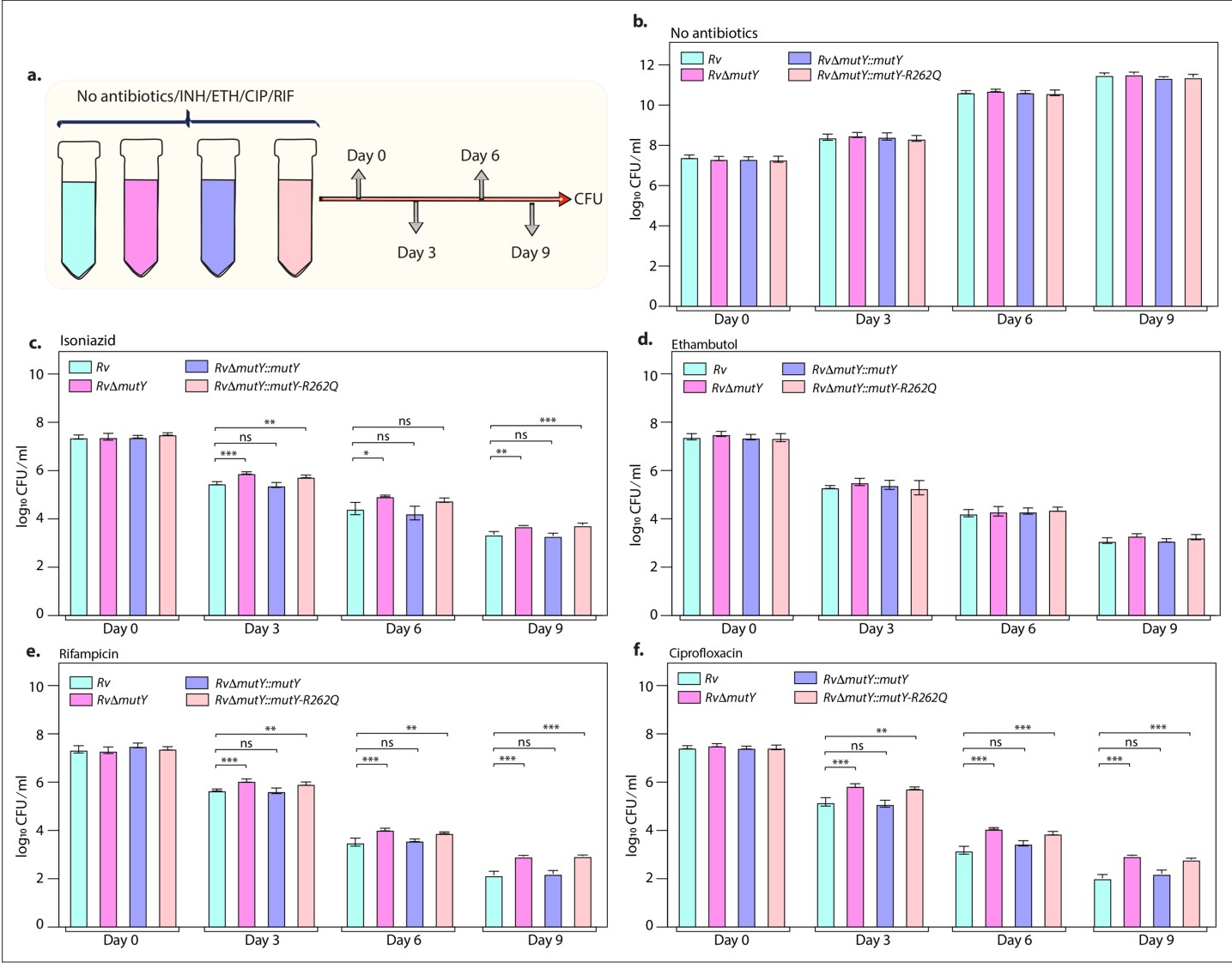

**Figure 4.** Killing kinetics in the presence of antibiotics show better survival of RvΔmutY and RvΔmutY::mutY R262Q. (**a**) Schematic representation of killing kinetics. (**b**) Growth kinetics in the absence of drugs. (**c–f**) Growth kinetics in the presence of isoniazid, rifampicin, ciprofloxacin, and ethambutol. Two biologically independent sets of experiments were performed. Each biological experiment was performed in biological triplicates. Data represent one set of experiments. Statistical analysis (two-way ANOVA) was performed using Graph pad prism software. ***p<0.0001, **p<0.001, and *p<0.01.

The online version of this article includes the following source data for figure 4:

**Source data 1.** Killing kinetics in the absence and presence of different antibiotics.

enumerated at 4 and 96 hr p.i. during the fourth (final) round of infection (*Figure 5a*). CFUs obtained at 4 hr p.i. showed equal load. *RvΔmutY* and *RvΔmutY::mutY*-R262Q exhibited better survival in the absence of antibiotics than *Rv* and *RvΔmutY::mutY* (*Figure 5b–e*). There was no additional advantage compared with untreated in the presence of isoniazid (*Figure 5f*). However, we observed a log-fold advantage for *RvΔmutY* and *RvΔmutY::mutY*-R262Q compared with *Rv* or *RvΔmutY::mutY* in the presence of rifampicin or ciprofloxacin (*Figure 5f*).

## Acquisition of direct target mutations ex vivo in the presence of drugs

We sought to determine if the improved survival of *mutY* mutant and *mutY* variant in the above experiment (*Figure 5*) is due to the acquisition of mutations in the direct target of antibiotics. To identify the mutations, we performed Whole Genome Sequencing (WGS). Genomic DNA extracted from 10 independent colonies (grown in vitro) was mixed in equal proportion prior to library preparation. Only

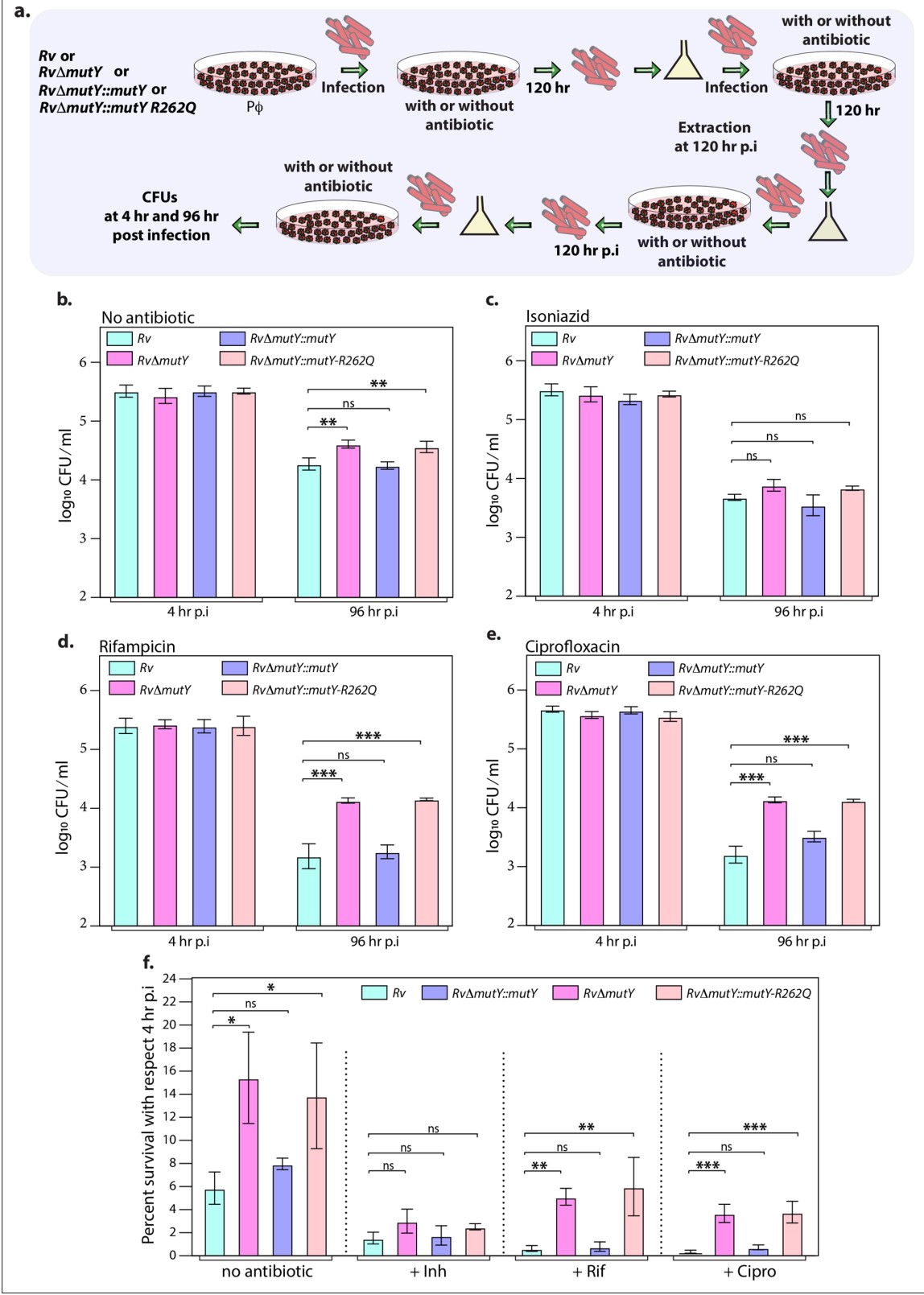

**Figure 5.** Mutations in the DNA repair genes provide a survival advantage in the presence of antibiotics. (**a**) A schematic is representing the ex vivo infection experiment in the presence and absence of different antibiotics. (**b–e**) Survival of the strains in the peritoneal macrophages at 4 and 96 hr post-infection (p.i.) without and with antibiotics (isoniazid or rifampicin, or ciprofloxacin). (**f**) Percent survival with respect to 4 hr p.i. was determined for each strain without and with antibiotics (isoniazid or rifampicin or ciprofloxacin). Two biologically independent sets of experiments were performed.

*Figure 5 continued on next page*

*Figure 5 continued*

Each biological experiment was performed in biological triplicates. Data represent one set of experiments. Statistical analysis (two-way ANOVA) was performed using Graph pad prism software. ***p<0.0001, **p<0.001, and *p<0.01.

The online version of this article includes the following source data and figure supplement(s) for figure 5:

**Source data 1.** Survival of different strains in the absence and presence of antibiotics ex vivo.

**Figure supplement 1.** Ex vivo survival of strains.

**Figure supplement 1—source data 1.** Survival of strains before and after passage in the peritoneal macrophages.

those SNPs present in >20% of the reads were retained for the analysis. Analysis of *Rv* sequences grown in vitro suggested that the laboratory strain accumulated 100 SNPs compared with the reference strain (data not shown). The sequence of the *Rv* laboratory strain was used as the reference for the subsequent analysis. WGS data for *RvΔmutY*, *RvΔmutY::mutY*, and *RvΔmutY::mutY*-R262Q strains grown in vitro did not show the presence of any mutations in the antibiotic target genes. In a similar vein, 10 independent colonies, each from the 7H11-OADC plates, after the final round of ex vivo infection in the presence or absence of antibiotics, were selected for WGS. Data indicated that in the absence of antibiotics, no direct target mutations were identified in the ex vivo passaged strains (***Figure 6a & e***). However, in the presence of isoniazid, we found mutations in the *katG* (Ser315Thr or Ser315Ileu) in the *Rv*, *RvΔmutY* but not in *RvΔmutY::mutY* and *RvΔmutY::mutY-R262Q* (***Figure 6b***

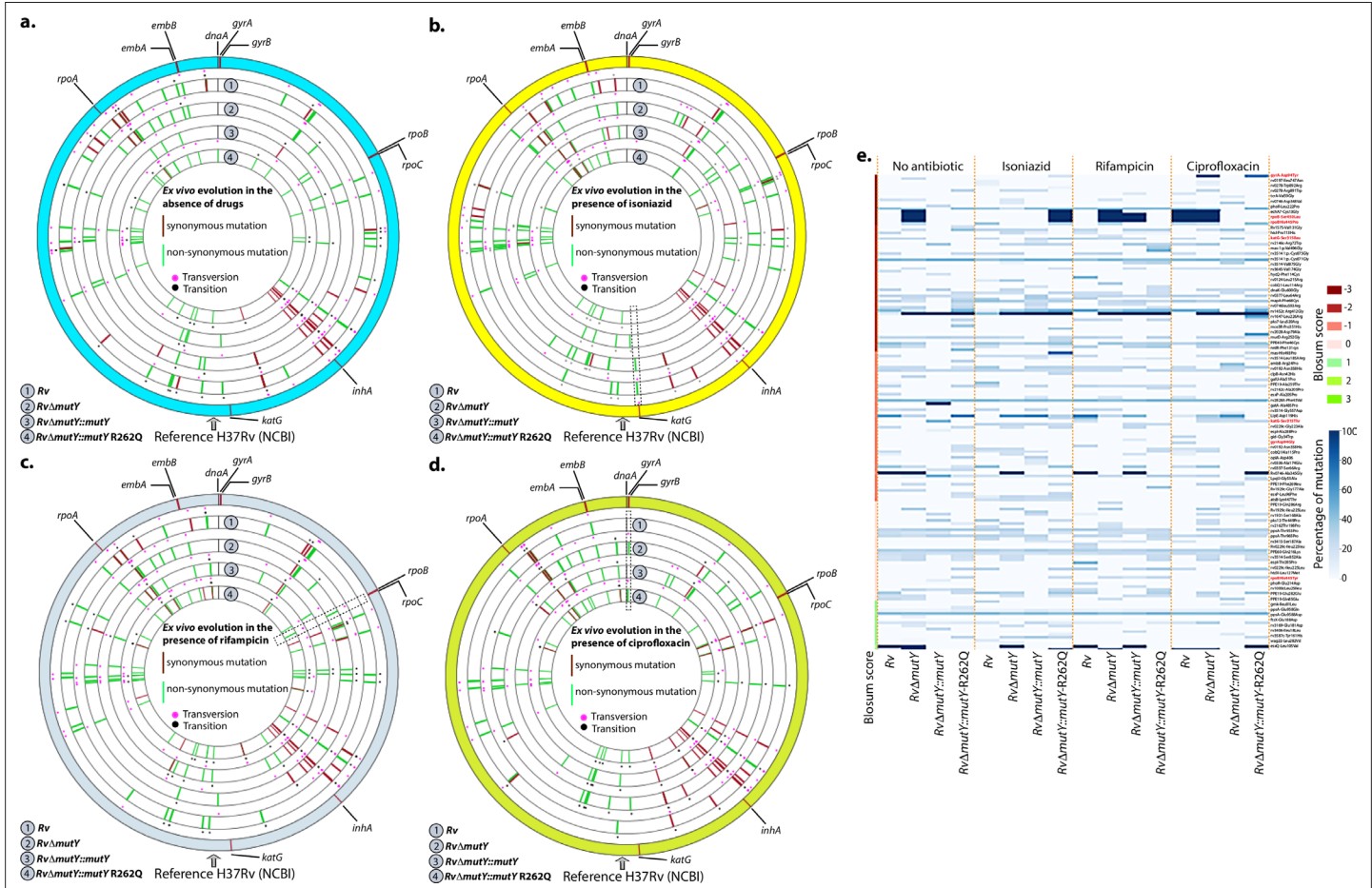

**Figure 6.** WGS reveals the acquisition of direct target mutations in the ex vivo passaged strains. (**a–d**) Circos plot showing the WGS analysis of the strains passaged ex vivo in the absence (**a**) and in the presence of isoniazid (**b**), rifampicin (**c**), and ciprofloxacin (**d**). The outermost circle represents the reference genome labeled with the known direct target mutations. Circles (from outside to inside) represent *Rv*, *RvΔmutY*, *RvΔmutY::mutY,* and *RvΔmutY::mutY*-R262Q genome. (**e**) Heat map represents the single nucleotide polymorphisms identified in the *Rv, RvΔmutY, RvΔmutY::mutY,* and *RvΔmutY::mutY*-R262Q after ex vivo passage.

*& e*). These findings are in congruence with the ex vivo evolution CFU analysis, wherein we did not observe a significant increase in the survival of *RvΔmutY* and *RvΔmutY::mutY*-R262Q in the presence of isoniazid (*Figure 5*). In the presence of ciprofloxacin and rifampicin, direct target mutations were identified in the *gyrA* and *rpoB* (*Figure 6c–e*). Asp94Glu/Asp94Gly mutations were identified in *gyrA*, and, His445Tyr/Ser450Leu mutations were identified in *rpoB* of *RvΔmutY* and *RvΔmutY::mutY*-R262Q, respectively. No direct target mutations were identified in the *Rv and RvΔmutY::mutY,* suggesting that the perturbed DNA repair aids in acquiring the drug resistance-conferring mutations in *Mtb* (*Figure 6c–e* & *Supplementary file 8*; *Paritosh, 2022*; https://github.com/kumar-paritosh/analysis_of_Mtb_genome).

## Competition experiment ex vivo reveals MutY variant confers a survival advantage

Results above suggested that the *mutY* variant impacts survival advantage when subjected to antibiotic selection, likely due to its ability to accumulate mutations. We reasoned that, if this is indeed the case, the *mutY* variant may outcompete the wild-type *Rv* when both the strains are present together. To test this hypothesis, we infected peritoneal macrophages with a combination *of Rv + RvΔmutY* or *Rv + RvΔmutY::mutY* or *Rv + RvΔmutY::mutY*-R262Q. At 96 hr p.i. host cells were lysed, and *Mtb* CFUs were enumerated on plain 7H11 or kanamycin (kan) (*Rv*) or hygromycin (hyg) (*RvΔmutY, RvΔmutY::mutY,* and *RvΔmutY::mutY*-R262Q) to evaluate the survival rates (*Figure 5—figure supplement 1c–d*). The percent survival was calculated as (CFUs obtained on kan or hyg/CFUs on kan + hyg) × 100. It is apparent from the data that the survival rates of competing strains were comparable, suggesting that *mutY* deletion or complementation with variant did not confer a significant advantage (*Figure 5—figure supplement 1c*). These results indicate that in the absence of prior antibiotic selection, deletion or the presence of *mutY* variant does not confer an advantage (*Figure 5—figure supplement 1c–d*).

To test these conclusions, we performed ex vivo co-infection experiment with the strains that were subjected to three rounds of prior selection (*Figure 5*). Peritoneal macrophages were infected with a combination *of Rv + RvΔmutY* or *Rv + RvΔmutY::mutY* or *Rv + RvΔmutY::mutY*-R262Q (*Figure 7a*). At 24 hr p.i., cells were either treated or not treated with an antibiotic for the subsequent 72 hr, and total CFUs were enumerated as described above to evaluate the survival rates (*Figure 5—figure supplement 1e–f*). As expected, there was no difference in the CFUs either at 4 hr p.i. (*Figure 7b–e*) or 24 hr p.i. (data not shown). At 96 hr p.i., *RvΔmutY* and *RvΔmutY::mutY*-R262Q strains showed a distinct advantage over *Rv* both in the absence or presence of antibiotics. Importantly, *RvΔmutY::mutY* did not show any advantage over *Rv* under any conditions. These results suggest that subjecting deletion or variant strains to antibiotic stress in the host helps in evolution of the strains that can outcompete the wild-type strain.

## *MutY* variant exhibits enhanced survival in vivo

Upon entering the host macrophages, *Mtb* encounters multiple forms of stress that impede its growth. To survive and grow in such a hostile environment, *Mtb* employs various defense mechanisms (*Chai et al., 2020*). The treatment regimen with anti-TB drugs imposes a supplementary layer of stress on the pathogen. An auxiliary mechanism the pathogen uses is to accumulate mutations in its genome that improve its ability to combat antibiotic and host-induced stresses. We asked if the variant mutations identified in DNA repair genes provide one such auxiliary mechanism. To test this hypothesis, we performed guinea pig infection experiments using *Rv, RvΔmutY, RvΔmutY::mutY,* and *RvΔmutY::mutY*-R262Q (*Figure 8a*). CFUs were enumerated after 1 and 56 days post-infection. CFUs obtained on day 1 showed the deposition was equal for wild-type, mutant, and the complemented strains. Gross pathology and histopathology analysis of infected lungs showed the presence of well-formed granulomas (*Figure 8b–c*). Significantly, 56 hr p.i., *RvΔmutY, RvΔmutY::mutY*-R262Q strains showed ~five-fold superior survival than *Rv RvΔmutY::mutY,* suggesting that the variant mutant identified indeed confers advantage (*Figure 8d*).

Next, we determined the survival ability of *RvΔmutY, RvΔmutY::mutY,* and *RvΔmutY::mutY*-R262Q, when competed against wild-type *Rv* strain (*Figure 8e*). The lung and spleen homogenates were plated on 7H11 to determine total CFUs (*Figure 8f*, *Figure 8—figure supplement 1*). The total CFUs were found to be comparable in both lungs and spleen across all combinations (*Figure 8f*).

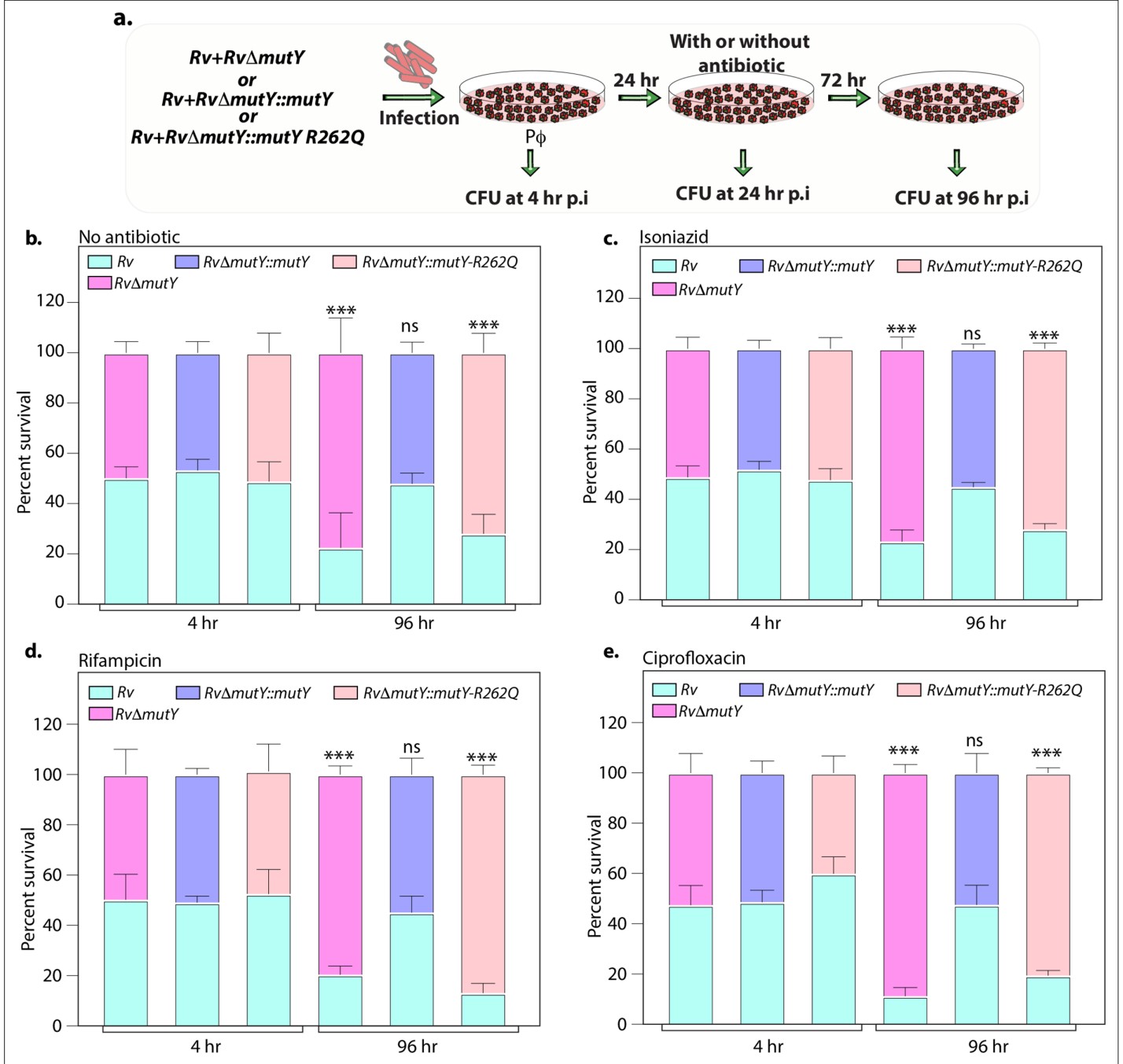

**Figure 7.** Variant of mutY outcompetes Rv in competition experiment. (**a**) Schematic representing the competition experiment performed in peritoneal macrophages. Strains obtained after three rounds of infection in the peritoneal macrophages were used to perform a competition experiment (*Figure 4a*). (**b–e**) Percent survival of *Rv, RvΔmutY, RvΔmutY::mutY,* and *RvΔmutY::mutY R262Q* in the absence and presence of antibiotics. Two biologically independent experiments, with each experiment performed in technical triplicates. Data represent one of the two biological experiments. Data represent mean and standard deviation. Statistical analysis (two-way ANOVA) was performed using Graph pad prism software. ***p<0.0001, **p<0.001, and *p<0.01.

The online version of this article includes the following source data for figure 7:

**Source data 1.** Competition experiment in the presence and absence of different drugs after passage in the peritoneal macrophages.

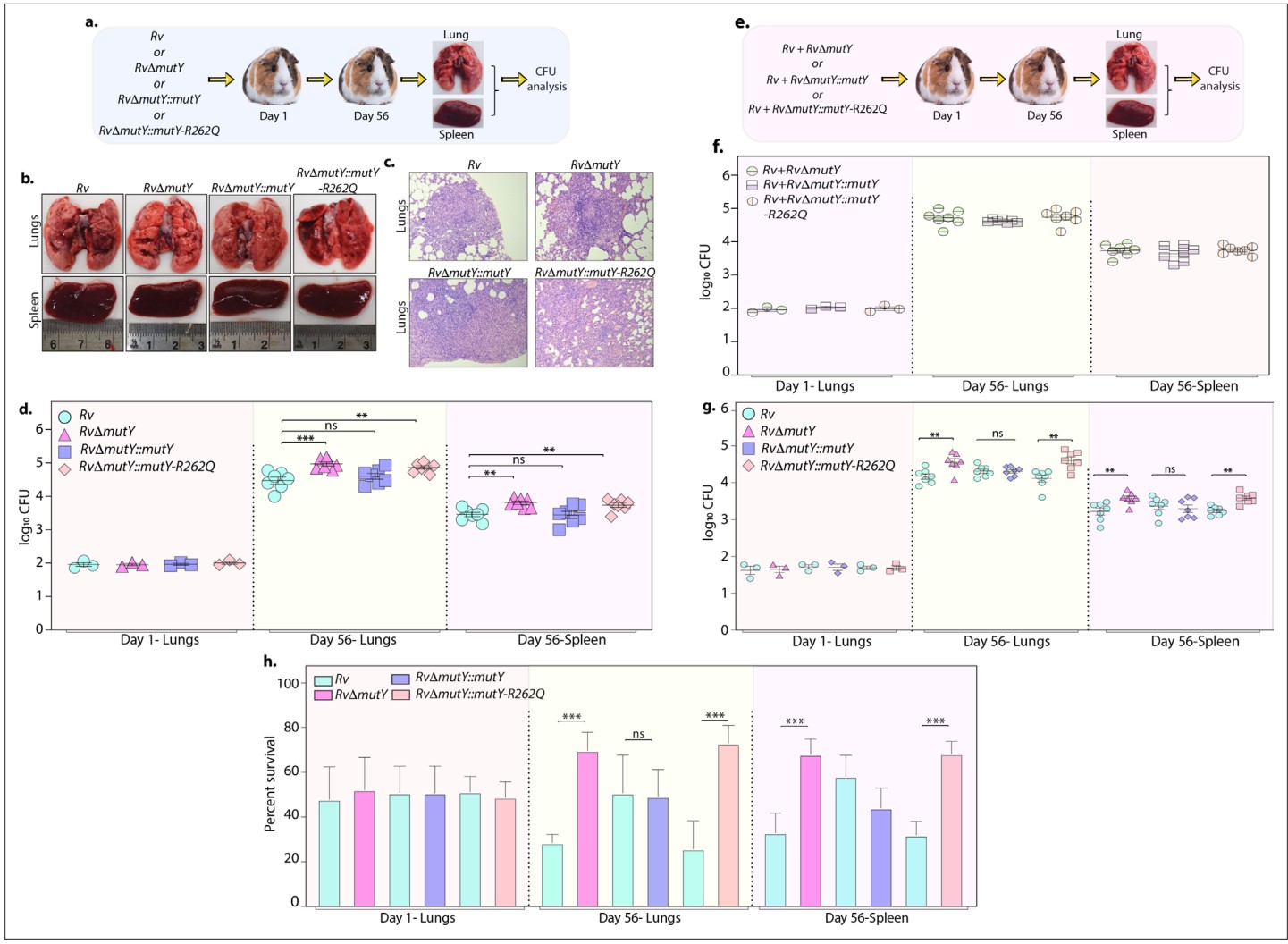

**Figure 8.** Perturbation of DNA repair results in enhanced survival in vivo. (**a**) A schematic representation of the guinea pig infection experiment. (**b**) Gross histopathology of lungs and spleen of infected guinea pigs. (**c**) Hematoxylin and eosin staining of infected lung tissue showing the well-formed granuloma. Magnification ×10. (**d**) Guinea pigs were challenged with *Rv, RvΔmutY, RvΔmutY::mutY,* and *RvΔmutY::mutY-R262Q* via the aerosol route. CFUs were enumerated at day 1 post-infection to determine the lungs' initial load (n=3). The CFUs at 1 day post-infection (p.i.) are represented for the whole lung. At 56 days p.i., lungs and spleen were isolated to determine survival (CFU / ml, n=7). (**e**) Outline showing the competition experiment performed in guinea pigs. (**f**) Total CFU enumeration of *Rv + RvΔmutY, Rv + RvΔmutY::mutY,* and *Rv + RvΔmutY::mutY-R262Q* in the lungs and spleen of guinea pigs at 1 day (whole lung, n=3) and 56 days p.i. (CFU/ml, n=7). (**g**) CFU enumeration of *Rv + RvΔmutY, Rv + RvΔmutY::mutY,* and *Rv + RvΔmutY::mutY-R262Q* on kanamycin and hygromycin containing plates. Statistical analysis was performed using two-way ANOVA. Graphpad prism was employed for performing statistical analysis. ***p<0.0001, **p<0.001, and *p<0.01. (**h**) Survival of each strain at indicated time points in the mixed infection. (**i**) Percent survival at 1 and 56 days in the competition experiment. Statistical analysis was performed using unpaired t-test. ***p<0.0001, **p<0.001, and *p<0.01.

The online version of this article includes the following source data and figure supplement(s) for figure 8:

**Source data 1.** Gross histopathology of the infected lungs and spleen isolated from guinea pig.

**Source data 2.** Haematoxylin and eosin staining.

**Source data 3.** Survival of different strains in vivo.

**Figure supplement 1.** Gross histopathology of infected lungs and spleen isolated at 56days post-infection.

**Figure supplement 1—source data 1.** Gross histopathology of lungs and spleen isolated from guinea pigs after competition experiment.

**Figure supplement 2.** WGS analysis of strains isolated from guinea pig lungs.

Lung and spleen homogenates were plated on either kan (*Rv*) or hyg (*RvΔmutY, RvΔmutY::mutY* or *RvΔmutY::mutY*-R262Q) containing plates to determine survival (*Figure 8g*). As with independent infections, *RvΔmutY,* or *RvΔmutY::mutY*-R262Q, showed fivefold elevated CFUs compared with *Rv* (*Figure 8g*). When plotted as percent survival, the data showed that *mutY* deletion or complementation with the variant confers a survival advantage to the pathogen (*Figure 8h*). To determine the plausible cause of enhanced survival in vivo, we performed WGS of the strain isolated from guinea pig lungs. Analysis revealed that the specific genes such as *cobQ1, smc, espI,* and *valS* were mutated only in *RvΔmutY and RvΔmutY::mutY*-R262Q but not in *Rv* and *RvΔmutY::mutY*. Besides, *tcrA* and *gatA* were mutated only in *RvΔmutY*, whereas *rv0746* was mutated exclusively in the *RvΔmutY::mutY* (*Figure 8—figure supplement 2a–b*). However, we did not observe any direct antibiotic resistance target mutations; this may be because guinea pigs were not subjected to antibiotic treatment. However, the total number of SNPs observed in all four strains was comparable. Thus the precise mechanistic role of MutY in *Mtb* pathogenesis needs further investigation. Collectively, results show that variants identified in DNA repair genes abrogate their function and contribute to a better survival in different stress conditions.

## Discussion

The WGS of clinical strains provides vital information about many aspects such as, the acquisition and transmission of drug resistance, evolution of compensatory mutations, and evolution of the drug resistance in patients (*Cohen et al., 2019*). A recent WGS analysis of 10,219 diverse *Mtb* isolates successfully predicted mutations associated with pyrazinamide resistance (*CRyPTIC Consortium and the 100,000 Genomes Project et al., 2018*). Large-scale GWAS, employed initially for analyzing human genome data, is an invaluable tool to delineate the mutations that confer antibiotic resistance in bacteria (*Power et al., 2017*). For example, using the PhyC test on 116 clinical strains of *Mtb*, *ponA1* was identified as one of the targets of independent mutations (*Farhat et al., 2013*). Analysis of 161 *Mtb* genomes identified polymorphisms in the intergenic regions that confer resistance to *p*-aminosalicylic acid (PAS) through overexpression of *thyA* and *thyX* (*Zhang et al., 2013*). GWAS of 498 sequences revealed that mutation in alanine dehydrogenase correlated with the resistance to a second-line drug D-cycloserine (*Desjardins et al., 2016*). Evaluation of 549 strains led to the identification of *prpR*, which in an ex vivo infection model confers conditional drug tolerance through regulation of propionate metabolism (*Hicks et al., 2018*). Largest GWAS involving 6465 clinical isolates uncovered novel resistance-associated mutations in *ethA* and *thyX* promoter, associated with ethionamide and PAS resistance, respectively (*Coll et al., 2018*).

We performed a gene-based GWAS analysis on a large dataset of susceptible and MDR/XDR clinical strains (*Figure 1*). We identified mutations in the known direct targets of both first- and second-line antibiotics and a few recently reported genetic variants using association analysis (*Tables 1 and 2*, *Table 3*). We also identified frameshift and non-synonymous mutations in *rv2333c* and *rv1250*, encoding for the transporters known to be differentially expressed in MDR patients (*Table 3*; *Umar et al., 2019*). Besides, the analysis captured mutations in genes involved in cell metabolism (*Figure 2*). This finding supports a recent study in *Escherichia coli,* where drug treatment led to the acquisition of mutations in the metabolic genes that impart drug resistance (*Lopatkin et al., 2021*). We believe that these may be compensatory mutations that placate the fitness cost associated with antibiotic resistance, and hence, might be the consequence of antibiotic resistance.

Most importantly, for the first time, we identified a significant association between mutations in three of the DNA repair pathway genes with drug resistance. This study is in line with the studies on the other pathogens such as *Pseudomonas aeruginosa*, *Helicobacter pylori*, *Neisseria*

**Table 3.** Transporters associated with drug resistance phenotype.

| Gene | False discovery rate-adjusted p-value | Variant |
|------|------|------|
| *rv1258c* | 1.60E-08 | Non-syn |
| *mmpL2* | 9.51E-05 | Non-syn |
| *rv0987* | 1.60E-08 | Non-syn |
| *rv1250* | 3.44E-07 | Non-syn |
| *kdpC* | 2.15E-07 | Non-syn |
| *rv0928* | 2.15E-07 | Non-syn |
| *rv2333c* | 1.32E-20 | Frameshift |
| *mmpL13a* | 1.25E-06 | Syn |
| *kdpB* | 2.15E-07 | Syn |

*meningitides,* and *Salmonella typhimurium*, where mutations or deletions in the DNA repair genes were identified in the clinical isolates (*Chopra et al., 2003*). Moreover, the deletion of *ung* and *udgB* (BER pathway genes), independently or together, provides a survival advantage to the bacteria (*Naz et al., 2021*). It is known that lineage 2 clinical strains have polymorphisms in the DNA repair and replication genes (*Mestre et al., 2011*; *Ebrahimi-Rad et al., 2003*). However, we identified mutations in DNA repair genes in lineage 4, suggesting that the phenomenon is not confined to lineage 2 (*Figure 3—figure supplement 1a*).

The recently published CRYPTIC consortium associates 12,289 Mtb strains with the resistance phenotype to 13 antibiotics does not contain a mutY-Arg262Gln mutation in their data set. This may be due to the different strains used in the study (*CRyPTIC Consortium and the 100,000 Genomes Project et al., 2018*). Mutation spectrum analysis of the clinical strains harboring the *mutY* variant and closely related strains showed a trend toward higher C→T, A→G, and C→A mutations, but we could not perform statistical analysis, as the strains harboring the *mutY* variant were limited. We investigated if the mutations in DNA repair genes are the cause or the consequence of the antibiotic resistance. We functionally validated the identified mutations of two different pathway genes, *mutY* and *uvrB*, in *Msm* using gene replacement mutants. Mutation frequency analysis suggests that GWAS identified Arg262Gln and Ala524Val mutations in *mutY* and *uvrB*, respectively, abrogated their functions (*Figure 3*). The data agrees with the previously published study, wherein WGS analysis of antibiotic susceptible strain isolated from a patient showed a non-synonymous mutation in UvrB (A582V). Notably, the variant strain evolved into XDR-TB over 3.5 years after the first- and second-line drug treatment (*Eldholm et al., 2014*). Therefore, we propose that mutations identified in DNA repair genes contribute to the evolution of antibiotic resistance (*Figure 2—figure supplement 5*). Besides, the killing kinetics in the presence of different anti-TB drugs show that *RvΔmutY* and *RvΔmutY::mutY*-R262Q exhibit better survival (*Figure 4*).

Poor adherence of the patients to the antibiotic regimen is the leading cause of the emergence of drug resistance. The continual treatment with antibiotics provides sufficient time to evolve strains into MDR or XDR. We emulated the condition by repeatedly exposing strains to antibiotics in the ex vivo model, followed by growing them in vitro without antibiotics. This led to the improved survival of *RvΔmutY* and *RvΔmutY::mutY*-R262Q (*Figure 5*). WGS of the strains after ex vivo passage shows that the compromised DNA repair helps in acquiring mutations in the direct targets of the antibiotics (*Figure 6*). We then determined the survival of ex vivo evolved strains in the mixed infection scenario. A competition experiment using the evolved strains showed that *RvΔmutY* and *RvΔmutY::mutY*-R262Q

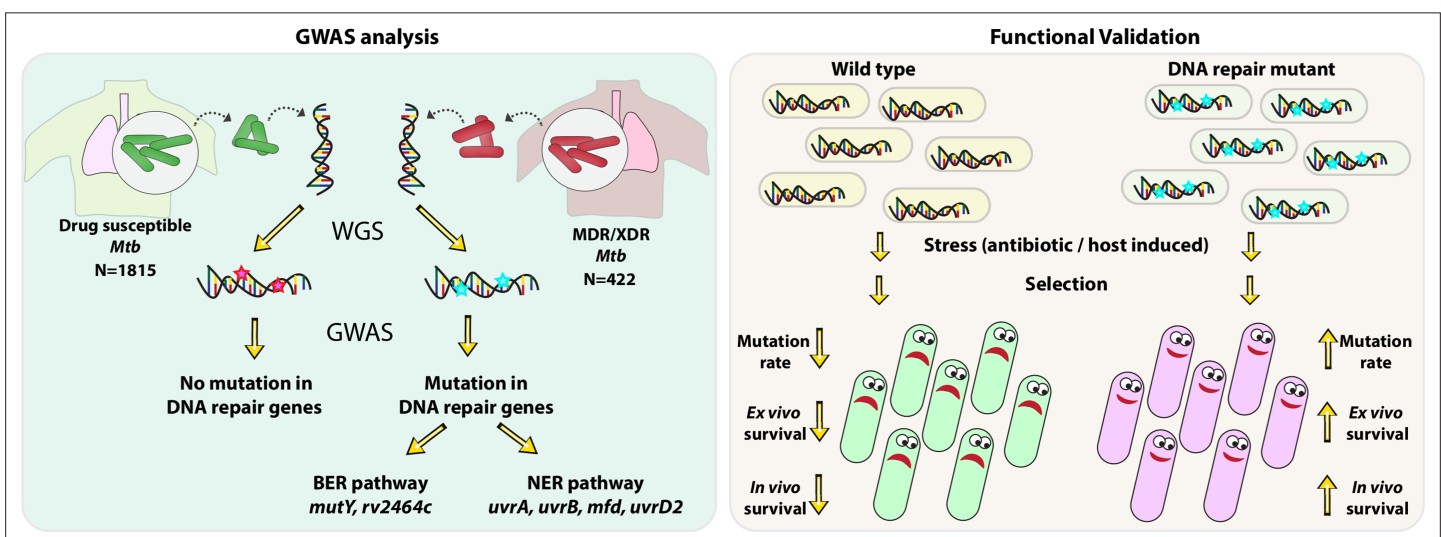

**Figure 9.** Model. Model depicts the analysis and subsequent validation. Genome-wide association study (GWAS) revealed mutation in three DNA repair pathway genes in multidrug resistant/extensively drug resistant (MDR/XDR) strains. Based on GWAS, we proposed that mutations in DNA repair genes are associated with the evolution of antibiotic resistance in *Mycobacterium tuberculosis* (Mtb). Functional validation was performed using the gene replacement mutants of base excision repair (BER) and nucleotide excision repair (NER) pathway genes in *Mycobacterium smegmatis* and Mtb. In vitro, ex vivo, and in vivo experiments show that compromised DNA repair pathway leads to the enhanced survival of bacteria.

could outcompete *Rv* (**Figure 7**). Host stress drives the selection of the bacterial population that acquires the ability to withstand adverse conditions. We evaluated the survival of the different strains in guinea pigs. Results suggest that *RvΔmutY* and *RvΔmutY::mutY*-R262Q exhibit improved survival. Similarly, we observed that *RvΔmutY* and *RvΔmutY::mutY*-R262Q could successfully outcompete *Rv* in the guinea pig infection model (**Figure 8**).

Using the GWAS approach and functional validation of the clinical mutations identified in the BER and NER pathways, we established a novel link between the compromised DNA repair and the evolution of antibiotic resistance (**Figure 9**).

Collectively, the data presented here suggest that the loss of function of DNA repair genes helps acquire drug resistance in the presence of anti-TB drugs. We propose that the evolution of MDR or XDR-TB is likely a consequence of the loss of function of DNA repair genes. The presence of mutations in DNA repair genes can be an early-stage diagnostic marker for the evolution of the strain into MDR/XDR-TB. Molecular diagnosis of DNA repair gene mutations at the onset of infection should help design better therapies to impede the evolution of these strains into MDR or XDR. These findings indicate that bacteria having compromised DNA repair can contribute to the accumulation of mutations, providing an advantage to the bacilli when subjected to antibiotic treatment.

# Materials and methods

## Sequence retrieval and variant calling

Accession numbers for 2773 clinical strains of *Mtb* were obtained from 10 previous studies (**Supplementary file 1**), representing all 4 lineages from 9 countries. Sequence data were retrieved from EBA (https://www.ebi.ac.uk/) and NCBI databases (https://www.ncbi.nlm.nih.gov/) and quality filtered using Trimmometic software (**Bolger et al., 2014**). Adapter and *E. coli* sequence contaminations were removed, the sequences were analyzed at a sliding window of 4 bp, and those with an average phread value of 15 were clipped. Parameters for the trimmometic-based QC were set as (Leading:3, slidingwindow:4:15 trailing:3 minlen:60). Reads with a filtered length of <60 bp were removed. The filtered SE/PE reads were mapped on the H37Rv reference genome (Accession number –ASM19595) using the Burrows Wheeler Alignment (BWA) mem algorithm (**Li and Durbin, 2009**). GATK pipeline was used for sorting, PCR duplicate removal, and realignment of sequences (**Van der Auwera et al., 2013**). Variants (SNP and small InDels) were predicted in a batch mode with Platypus script (**Rimmer et al., 2014**). SNPs with read depth <5 or mapping quality of <20 were marked as missing. SNPs with missing calls in >40% of the accessions were removed from the analysis. Finally, ~160,000 SNPs from 2773 accessions with a variant dataset with an MAF of >1% were selected for the final analysis. The.vcf file was converted to.hmp file with Tassel (**Bradbury et al., 2007**). A phylogenetic tree was generated using ~160,000 SNPs using SNPhylo (**Lee et al., 2014**).

## Association analysis

Genome-wide association analysis was carried out with 1815 susceptible strains and 422 MDR/XDR strains (**Supplementary file 2**). A GAPIT based on a compressed mix linear model was performed under the R environment (**Lipka et al., 2012**). VanRaden algorithm was employed for the calculation of the Kinship matrix. The kinship matrix assessed the relatedness among the strains included in the association panel. Principal components in GAPIT were used to classify the population structure. An association mapping analysis was carried out by combining the population structure analysis and the relative kinship matrix. The association mapping analysis obtained p-values, R2, and marker effect values. The FDR adjusted p-values in the GAPIT software are highly stringent as it corrects the effects of each marker based on the population structure (Q) as well as kinship (K) values and often lead to overcorrection (**Gao et al., 2016**; **Zegeye et al., 2014**). We selected a corrected p-value of $10^{-5}$ as the threshold for selecting associated genes. The associated SNPs were annotated using the snpEff v4.11 (**Cingolani et al., 2012**). A snpeff database was generated using the H37Rv (ASM19595v2, was used as a reference for mapping). The corresponding .gff file and the SNPs were annotated based on their position on the genome. Dot-plot, Manhattan's plot, and volcano plot were generated using R scripts.

## Generation of gene replacement mutant in *Mtb*

The upstream (5' flank) and downstream region (3' flank) of the *mutY* were amplified using the *Rv* genomic DNA. Flanks were digested with an appropriate restriction enzyme and ligated with oriE + lambda cos and hygromycin resistance cassette (*Jain et al., 2014*). The allelic exchange substrate was digested to generate linearized substrate and electroporated in the recombineering proficient *Rv* strain harboring pNit-ET plasmid (*van Kessel and Hatfull, 2007*). Colonies were screened post 3 weeks electroporation for gene replacement mutant.

## Generation of complementation constructs and western blot analysis

Wild type allele of *mutY* or *uvrB* was PCR amplified using *Rv* genomic DNA. The PCR product and the vector pSTL-giles were digested with NdeI and HindIII to generate pSTL-giles-*mutY* or pSTL-giles-*uvrB*. Subsequently, the PCR product was ligated with the vector pSTL-giles. SapI-based cloning was employed for the generation of pSTL-giles-*mutY-R262Q* or pSTL-giles-*uvrB-A524V*. Constructs, pSTL-giles-*mutY* and pSTL-giles-*mutY-R262Q,* were electroporated in the *msmΔmutY* or *RvΔmutY* to generate *msmΔmutY::mutY* and *msmΔmutY::mutY-R262Q* or *RvΔmutY::mutY* and *RvΔmutY::mutY-R262Q*. Constructs, pSTL-giles-*uvrB* or pSTL-giles-*uvrB-A524V,* were electroporated in *msmΔuvrB* to generate *msmΔuvrB::uvrB* or *msmΔuvrB::uvrB- A524V* (oligonucleotides used in the study are given in the *Supplementary file 9*). 50 ml cultures of *Rv*, *RvΔmutY*, *RvΔmutY::mutY,* and *RvΔmutY::mutY-R262Q* were inoculated in 7H9-ADC medium at $A_{600}$ ~0.1 and grown till $A_{600}$ ~0.8. Cells were pelleted in 50-ml falcon tubes at 4000 rpm for 10 min and resuspended in the lysis buffer containing protease inhibitors. Cells were transferred in the bead beating tubes containing zirconium beads. Bead beating was performed for eight cycles. The cell lysate was centrifuged twice at 13,000 rpm for 45 min at 4°C. Protein was estimated using the Bradford assay reagent. 50 μg of *Rv*, *RvΔmutY*, *RvΔmutY::mutY,* and *RvΔmutY::mutY-R262Q* was loaded on two independent 10% SDS-PAGE and transferred to nitro-cellulose membrane. 5% BSA prepared in $1XPBST_{20}$ was used for blocking the membrane for 2 hr. Membranes were incubated overnight at 4°C with α-FLAG (1:5000), andα-GroEL-1 (1:10000), respectively. Membranes were washed using $1XPBST_{20}$ (thrice) and incubated with anti-rabbit secondary antibody DARPO (1:10000) for 1 hr at room temperature (RT; 25°C). Membranes were washed thrice with $1XPBST_{20,}$ and a blot was developed using a chemiluminescence reagent.

## Analysis of mutation frequency and rate

Antibiotic sensitive cultures of *msm*, *msmΔmutY*, *msmΔmutY::mutY*, *msmΔmutY::mutY-R262Q*, *msmΔuvrB*, *msmΔuvrB::uvrB,* and *msmΔuvrB::uvrB-A524V* were grown in 7H9-ADC medium $A_{600}$ ~0.6, and 50,000 cells/ml were inoculated in fresh 10-ml medium. Cultures were grown for 6 days in a 37°C incubator at 200 pm. On the seventh day, appropriate dilutions were plated on 7H11-OADC plain plates to determine the load, and 1 ml was plated on rifampicin (50 μg/ml). *Rv*, *RvΔmutY*, *RvΔmutY::mutY,* and *RvΔmutY::mutY* R262Q strains were inoculated at $A_{600}$~0.1. Strains were grown to $A_{600}$ ~0.6 and plated on 7H11-plain or rifampicin (2 μg/ml) or isoniazid (5 μg/ml) plates to calculate the mutation frequency. Six antibiotic-sensitive colonies were grown in a 7H9-ADC medium up to $A_{600}$ ~0.8, and 50,000 cells/ml were inoculated in a fresh 10-ml 7H9-ADC medium in the presence of 15% sterile *Rv* culture filtrate. On the 15th day, appropriate dilutions were plated on 7H11-OADC plain plates to determine the load, and 1 ml was plated on rifampicin (2 μg/ml), isoniazid (5 μg/ml), and ciprofloxacin (1.5 μg/ml). The mutation rate was determined as reported previously (*Boshoff et al., 2003*; *David, 1970*). Two biologically independent experiment sets were performed to determine mutation frequency. Each experiment was performed in the biological triplicate (mutation frequency) or six triplicate (mutation rate). Data represent one of the two biological sets of experiments. Data represent mean and standard deviation. Statistical analysis (two-way ANOVA) was performed using Graphpad prism software. ***p<0.0001, **p<0.001, and *p<0.01.

## Killing kinetics in the presence of anti-TB drugs

*Rv*, *RvΔmutY*, *RvΔmutY::mutY,* and *RvΔmutY::mutY*-R262Q were grown in 7H9-ADC medium up to $A_{600}$ ~0.8. Cultures were inoculated in the fresh medium at the $A_{600}$ ~0.6 in 10-ml 7H9-ADC medium. Different antibiotics- rifampicin (2 μg/ml), isoniazid (5 μg/ml), and ciprofloxacin (1.5 μg/ml) were added, and the CFUs were enumerated at day 0, 3, 6, and 9 post on 7H11-OADC plates. Two biologically independent sets of experiments were performed. Each experiment was performed in the biological

triplicate. Data represent one of the two biological sets of experiments. Data represent mean and standard deviation. Statistical analysis (two-way ANOVA) was performed using Graphpad prism software. ***p<0.0001, **p<0.001, and *p<0.01.

## Survival of strains ex vivo

Balb/c mice were injected with thioglycollate, and 72 hr post-injection, peritoneal macrophages were isolated. One million cells were seeded in each well of a six-well plate. Cells were infected with *Rv*, *RvΔmutY*, *RvΔmutY::mutY,* and *RvΔmutY::mutY* R262Q independently at an multiplicity of infection (MOI) of 1:5. After 24 hr p.i. cells were treated with rifampicin (1 μg/ml), isoniazid (1 μg/ml), and ciprofloxacin (2.5 μg/ml). Cells were lysed at 120 hr p.i. using 0.05% sodium dodecyl sulfate (SDS), and the bacteria was extracted. Bacteria were washed thrice with PBS to ensure the removal of SDS. Extracted bacteria were cultured in a 7H9-ADC medium without antibiotics for 5–7 days up to $A_{600}$~0.4. These cultures were used for the next round of infection. The whole process was repeated three times. During the fourth round of infection, CFUs were enumerated at 4 hr p.i. and 96 hr p.i. to determine the survival of different strains. Percent survival was calculated by normalized CFU obtained at 96 hr with respect to the respective CFUs obtained at 4 hr. Two biologically independent experiment sets were performed. Each experiment was performed in biological triplicate. A representative experiment is shown in *Figure 5*. Similarly, the strains obtained after three rounds of infection from the above experiment were used for the competition experiment. *Rv + RvΔmutY*, *Rv + RvΔmutY::mutY,* and *Rv + RvΔmutY::mutY-R262Q* were mixed in a 1:1 ratio. 24 hr p.i, the cells were either not treated or treated with the antibiotics (isoniazid-1μg/ml, rifampicin- 1 μg/ml, and ciprofloxacin- 2.5 μg/ml) for subsequent 72 hr. CFUs were enumerated at 4 hr and 96 hr p.i. on 7H11-plain (all strains), 7H11-Kan (Rv), and 7H11-Hyg (*RvΔmutY*, *RvΔmutY::mutY*, and *RvΔmutY::mutY-R262Q*) plates. Percent survival was calculated as (CFUs on Kan or Hyg plates/[CFUs on Kan +CFUs on Hyg])×100. Two biologically independent sets of experiments were performed. Each experiment was performed in the biological triplicate. Data represent one of the two biological sets of experiments. Statistical analysis (two-way ANOVA) was performed using Graph pad prism software. ***p<0.0001, **p<0.001, and *p<0.01.

## WGS of strains under different conditions

Independent colonies of *Rv* (n=10), *RvΔmutY* (n=10), *RvΔmutY::mutY* (n=10), and *RvΔmutY::mutY-R262Q* (n=10) were randomly selected and grown in vitro without any selection for genomic DNA isolation. *Rv* (n=10), *RvΔmutY* (n=10), *RvΔmutY::mutY* (n=10), and *RvΔmutY::mutY-R262Q* (n=10) were obtained after ex vivo passage in the absence and in the presence of isoniazid, rifampicin, and ciprofloxacin were grown independently for genomic DNA isolation. Similarly, *Rv* (n=10), *RvΔmutY* (n=10), *RvΔmutY::mutY* (n=10), and *RvΔmutY::mutY-R262Q* (n=10) isolated from guinea pig lungs were grown for the genomic DNA isolation. During the library preparation, genomic DNA isolated from an independent colony of *Rv* was mixed in equal amounts. The library of the mixed sample was prepared, and the library was sent for WGS after a quality check. A similar procedure was used to prepare each strain's libraries under different conditions. Variant calling was performed as described in the section-sequence retrieval and variant calling. SNPs less than or equal to 20% cut-off were discarded. A final matrix containing identified SNPs in the genes, mutation percentage, and blosum score was generated. Heat maps and circos plots were generated using custom python scripts.

## Guinea pig infection

*Rv*, *RvΔmutY*, *RvΔmutY::mutY*, and *RvΔmutY::mutY-R262Q* cultures were grown up to $A_{600}$ ~0.8. For preparing single-cell suspension, *Rv*, *RvΔmutY*, *RvΔmutY::mutY*, and *RvΔmutY::mutY-R262Q* cultures were pelleted at 4000 rpm at RT. After suspending cells in saline, cells were passed through a 26½ gauge needle to obtain a single-cell suspension. $1×10^8$ cells were taken in the 15 ml saline for infection. Female outbred Hartley guinea pigs were challenged using a Madison chamber calibrated to deliver ~100 bacilli/lung through the aersolic route. For determining the deposition of bacteria in the lungs of guinea pigs (n=3), CFUs were enumerated at 1 day post-infection on 7H11-plain plates for each strain. Survival of each strain was determined at 56 days post-infection in the lungs and spleen (n=7). *Rv + RvΔmutY*, *Rv + RvΔmutY::mutY,* and *Rv + RvΔmutY::mutY* R262Q were mixed in a 1:1 ratio, and the guinea pigs (n=10 per strain) were challenged as described above. CFUs were enumerated at 1 and 56 days post-infection on 7H11-plain plates or those containing kanamycin

or hygromycin-containing plates. Data represent the SEM. Statistical analysis (two-way ANOVA) was performed using Graph pad prism software. ***p<0.0001, **p<0.001, and *p<0.01. Percent survival was calculated as (CFUs on Kan or Hyg plates/[CFUs on Kan + CFUs on Hyg])×100. Data represents Standard deviation and mean. Statistical analysis (Unpaired t-test) was performed using Graph pad prism software. ***p<0.0001, **p<0.001, and *p<0.01. All guinea pig infection experiments were performed at the same time. Guinea pigs were not treated with any antibiotics before or after infection.

## Acknowledgements

This work was funded by the Department of Biotechnology, Government of India (BT/PR13522/COE/34/27/2015) and the J.C Bose fellowship (JCB/2019/000015). SN is a Senior Project Associate in the J.C Bose fellowship (JCB/2019/000015). We thank the Tuberculosis Aerosol Challenge Facility at ICGEB and staff for their help in performing animal infection experiments. We are thankful to the bio-containment facility (BSL3) at NII. Vector-pYUB1471 is a kind gift from Prof. William R Jacobs's laboratory.

## Additional information

### Funding

| Funder | Grant reference number | Author |
| --- | --- | --- |
| Department of Biotechnology, Ministry of Science and Technology, India | BT/PR13522/COE/34/27/2015 | Vinay Kumar Nandicoori |
| Department of Science and Technology, Ministry of Science and Technology, India | JCB/2019/000015 | Vinay Kumar Nandicoori |

The funders had no role in study design, data collection and interpretation, or the decision to submit the work for publication.

### Author contributions

Saba Naz, Conceptualization, Validation, Investigation, Visualization, Methodology, Writing - original draft, Writing - review and editing; Kumar Paritosh, Conceptualization, Data curation, Software, Formal analysis; Priyadarshini Sanyal, Sidra Khan, Investigation; Yogendra Singh, Supervision; Umesh Varshney, Resources, Supervision; Vinay Kumar Nandicoori, Conceptualization, Resources, Supervision, Funding acquisition, Writing - original draft, Project administration, Writing - review and editing

### Author ORCIDs

Saba Naz http://orcid.org/0000-0002-9901-4319
Yogendra Singh http://orcid.org/0000-0002-3902-4355
Umesh Varshney http://orcid.org/0000-0003-3196-5908
Vinay Kumar Nandicoori http://orcid.org/0000-0002-5682-4178

### Ethics

Animal experiments protocol was approved by the Animal Ethics Committee of the National Institute of Immunology, New Delhi, India. The approval (IAEC#409/16) is as per the guidelines issued by the Committee for the Purpose of Control and Supervision of Experiments on Animals (CPCSEA), Government of India.

### Decision letter and Author response

Decision letter https://doi.org/10.7554/eLife.75860.sa1
Author response https://doi.org/10.7554/eLife.75860.sa2

## Additional files

### Supplementary files

• Supplementary file 1. Total number of clinical strains used in this study. The table contains the total number of clinical strains obtained from different studies.

• Supplementary file 2. Clinical strains used for the Genome-wide association analysis. The table contains the clinical strains which are used for performing genome-wide association study analysis.

• Supplementary file 3. Synonymous change identified in the association analysis. The table contains synonymous changes identified in the multidrug-resistant/extensively drug-resistant strains.

• Supplementary file 4. Non-synonymous change identified in the association analysis. The table contains non-synonymous changes identified in the multidrug-resistant/extensively drug-resistant strains.

• Supplementary file 5. Upstream gene variants identified in the association analysis. The table contains non-upstream gene variants identified in the multidrug-resistant/extensively drug-resistant strains.

• Supplementary file 6. Stop codon or frameshift mutations identified in the association analysis. The table contains Stop codon or frameshift mutations identified in the multidrug-resistant/extensively drug-resistant strains.

• Supplementary file 7. Codon usage of the multidrug-resistant/extensively drug-resistant (MDR/XDR) strains. The table contains codon usage in the MDR/XDR and Rv strain.

• Supplementary file 8. Mutations identified in Rv, RvΔmutY, RvΔmutY::mutY, and RvΔmutY::mutY-R262Q strains under different conditions. The table consists of percentage of mutation and blosum score of genes that harbor s under different conditions.

• Supplementary file 9. Oligonucleotide used in the study. The table consists of oligonucleotide used in the study.

• Transparent reporting form

### Data availability

All data generated or analysed during this study are included in the manuscript and supporting file. Sequencing data have been deposited at NCBI under Bioproject PRJNA885615. Code availability at GitHub: https://github.com/kumar-paritosh/analysis_of_Mtb_genome; (copy archived at swh:1:rev:cf2547e50b00c57dee9b60bca899ed50e617106f).

The following dataset was generated:

| Author(s) | Year | Dataset title | Dataset URL | Database and Identifier |
|---|---|---|---|---|
| Nandicoori VK | 2022 | Whole genome sequencing of Mycobacterium tuberculosis strains under different conditions | http://www.ncbi.nlm.nih.gov/bioproject/?term=PRJNA885615 | NCBI BioProject, PRJNA885615 |

The following previously published datasets were used:

| Author(s) | Year | Dataset title | Dataset URL | Database and Identifier |
|---|---|---|---|---|
| Hicks ND, Yang J, Zhang X, Zhao B, Grad YH, Liu L | 2018 | Clinically prevalent mutations in Mycobacterium tuberculosis alter propionate metabolism and mediate multidrug tolerance | https://www.ncbi.nlm.nih.gov/bioproject/?term=PRJNA268900 | NCBI BioProject, PRJNA268900 |

*Continued on next page*

*Continued*

| Author(s) | Year | Dataset title | Dataset URL | Database and Identifier |
|---|---|---|---|---|
| Zhang H, Li D, Zhao L, Fleming J, Lin N, Wang T | 2013 | Genome sequencing of 161 Mycobacterium tuberculosis isolates from China identifies genes and intergenic regions associated with drug resistance | https://www.ncbi.nlm.nih.gov/sra?term=SRA065095 | NCBI Sequence Read Archive, SRA065095 |
| Casali N, Nikolayevskyy V, Balabanova Y, Harris SR, Ignatyeva O, Kontsevaya I | 2014 | Evolution and transmission of drug-resistant tuberculosis in a Russian population | https://www.ebi.ac.uk/ena/browser/view/PRJEB2138?show=reads | ERP000192, PRJEB2138 |
| Blouin Y, Hauck Y, Soler C, Fabre M, Vong R, Dehan C | 2012 | Significance of the identification in the Horn of Africa of an exceptionally deep branching Mycobacterium tuberculosis clade | https://www.ebi.ac.uk/ena/browser/view/PRJEB3334?show=reads | ERP001885, PRJEB3334 |
| Shanmugam S, Kumar N, Nair D, Natrajan M, Tripathy SP, Peacock SJ | 2018 | Genome Sequencing of Polydrug-, Multidrug-, and Extensively Drug-Resistant Mycobacterium tuberculosis Strains from South India | https://www.ncbi.nlm.nih.gov/bioproject/492975 | NCBI BioProject, 492975 |
| Guerra-Assuncao JA, Houben RM, Crampin AC, Mzembe T, Mallard K, Coll F | 2015 | Recurrence due to relapse or reinfection with Mycobacterium tuberculosis: a whole-genome sequencing approach in a large, population-based cohort with a high HIV infection prevalence and active follow-up | https://www.ebi.ac.uk/ena/browser/view/PRJEB2794?show=reads | ERP001072, PRJEB2794 |
| Clark TG, Mallard K, Coll F, Preston M, Assefa S, Harris D | 2013 | Elucidating emergence and transmission of multidrug-resistant tuberculosis in treatment experienced patients by whole genome sequencing | https://www.ebi.ac.uk/ena/browser/view/PRJEB2424 | EBI, PRJEB2424 |
| Bryant JM, Harris SR, Parkhill J, Dawson R, Diacon AH, van Helden P | 2013 | Whole-genome sequencing to establish relapse or re-infection with Mycobacterium tuberculosis: a retrospective observational study | https://ddbj.nig.ac.jp/resource/sra-submission/ERA020628 | DDBJ, ERA020628 |
| Walker TM, Cl IP, Harrell RH, Evans JT, Kapatai G, Dedicoat MJ | 2013 | Whole-genome sequencing to delineate Mycobacterium tuberculosis outbreaks: a retrospective observational study | https://www.ebi.ac.uk/ena/browser/view/PRJEB2221 | ERP000276, PRJEB2221 |

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
