## [Editor Report]

This paper provides important evidence implicating polymorphisms in the mycobacterial adenine DNA glycosylase, MutY, in the emergence of antibiotic resistance in *Mycobacterium tuberculosis*. While the precise mechanism underlying this phenotype requires further investigation, the inference from genome-wide association analyses of sequenced clinical isolates, supported by laboratory experiments and animal infection models, is convincing. This work adds a new locus of interest to the list of polymorphisms associated with tuberculosis drug resistance, and is likely to be relevant to the mycobacterial research field.

---

## [Decision Letter]

**Decision letter after peer review:**

Thank you for submitting your article "GWAS and functional studies implicate a role for altered DNA repair in the evolution of drug resistance in *Mycobacterium tuberculosis*" for consideration by *eLife*. Your article has been reviewed by 3 peer reviewers, one of whom is a member of our Board of Reviewing Editors, and the evaluation has been overseen by Gisela Storz as the Senior Editor. The reviewers have opted to remain anonymous.

The reviewers have discussed their reviews with one another, and the Reviewing Editor has drafted this summary to help you prepare a revised submission.

Essential revisions:

The reviewers were in general agreement that the work presents interesting and potentially important results; however, all reviewers felt that, in its current form, the conclusions are inadequately supported by the data. The following are therefore considered essential to address the major concerns:

1) The GWAS analysis provides the foundation of the paper, yet all three reviewers expressed concerns about the presentation of – and, therefore, inferences derived from – this part of the study. More, and clearer, information is required about the strains included, their potential phylogenetic relatedness, and the distributions of the identified resistance-associated mutations.

2) A key claim of this work is that the in vivo fitness advantage of the MutY mutant strain is due to hypermutagenesis and selection of a fitter strain, however the evidence supporting this conclusion is lacking.

3) Related to point (2) above: the clinical *M. tuberculosis* strains carrying the identified mutY and uvrB mutations should contain additional polymorphisms as a consequence of the loss/impairment of core DNA repair function(s), however no evidence is presented in support of (or refuting) this assumption.

4) There was some uncertainty among the reviewers whether the phenotypes reported are genuine drug-resistance, or rather a form of (heritable) tolerance. This uncertainty, which might be ascribed to the use throughout the manuscript of mutation frequency assessments, not mutation rates, must be resolved. For example, through the use of fluctuation assays; by determining resistance levels quantitatively (i.e., in MIC assessments); and/or by establishing the breadth of antimycobacterial agents (cidal and static drugs with different targets/mechanisms of action) to which reduced susceptibility is conferred by the identified DNA repair polymorphisms.

*Reviewer #1 (Recommendations for the authors):*

This manuscript presents intriguing data suggesting a role for defective DNA repair pathways in the development of drug resistance under treatment. However, there are a few key issues/questions which must be addressed.

(i) The GWAS analysis is confusing and/or unclear as described:

a. The authors report (L85-7) the use of known drug resistance phenotypes (where available) or inferences of drug-resistance from genotypic data to enhance the potential to identify other mutations that might be implicated in enabling the DR mutations, yet their list of known DR mutations (Table 1) seems predominantly to comprise rare or unusual mutations, not those commonly associated with clinical DR-TB.

b. In the same lines (85-7), the authors state that strains were separated into 5 drug susceptibility categories, the last of which was "pre-XDR", yet later sentences (e.g., L95, 104, 128, etc.) refer to "MDR/XDR" strains. Are these XDR or pre-XDR?

c. The distributions of the identified resistance-associated mutations across the different lineages need to be explained more clearly; this is necessary to strengthen claims of specific selection of the mutations under antibiotic treatment.

(ii) A central claim of the paper is that enhanced fitness, as a consequence of elevated mutagenesis, contributes to the prevalence of the uvrB and mutY mutations among the DR Mtb strains. While plausible, this is not explored in the manuscript: instead, the experimental work is limited to assessments of competitive survival in various models, with/without antibiotic selection, or to mutant frequency analyses. There is no direct evidence of increased mutational load in the mutY and uvrB strains. To strengthen the credibility of this central claim, WGS data of "successful"/DR mutants would be very useful.

(iii) Continuing from the above, it appears that these mutations confer heritable tolerance, rather than resistance. This does not undermine the value of this work, but it does suggest additional lines of experimental inquiry, for example:

a. mutation rate (not frequency) assessments;

b. use of higher drug concentrations (e.g., rifampicin at 200 ug/ml in M. smegmatis);

c. MIC assays to indicate precisely (quantitatively) the impact of the mutations on drug susceptibility;

d. expansion of the drug panel to include compounds with different mechanisms of action, cidal as well as static; that is, how broadly tolerant are these strains as a consequence of the individual mutations?

(iv) Continuing from the above, increased mutagenesis is generally deleterious, especially where enabled by the loss of core DNA repair functions; it would be useful to know, therefore, whether the mutY and uvrB mutations – both genes encoding proteins involved in excision repair pathways – incur any fitness costs. This could be assessed via DNA damage tolerance/survival assays. Also, are both mutations tolerated in a single strain (is there any evidence of this in the GWAS data?)? That is, do dual mutY uvrB alleles enhance drug survival, or reduce it, compared to the single mutants alone?

(v) Why is the wild-type parental strain (Rv) selectable on kanamycin-containing media? Presumably this is because it contains the complementation vector?

*Reviewer #3 (Recommendations for the authors):*

Text improvements:

– It is curious that this GWAS analysis did not identify some of the most common acquired drug-resistance conferring mutations. For example, as the authors note, this analysis identifies rpoB p.Leu452Pro (note that in the main text this is mislabeled as p.Leu452Val) and p.Val496Met, but misses the far more common Ser450L and Asp435Val alleles. A discussion as to why these (and other) known drug resistance mutations did not meet the cutoffs would be welcomed.

– Lines 49-50: "Although Mtb has a lower mutation rate…" Lower than what?

– Lines 56-59: "Despite well-known mechanisms of drug resistance, in 10-40% of the clinical isolates of Mtb, drug resistance cannot be determined by the mutations in the direct targets of antibiotics, implying the presence of hitherto unknown mechanisms that foster the development of resistance in Mtb (8)." It would be worth updating this statement based on the most recent work from the CRYPTIC consortium, for example.

– Lines 89-91: "The total number of SNPs observed for susceptible, mono-DR, MDR, or pre-XDR strains were comparable, suggesting no genetic drift during the evolution of antibiotic resistance (Figure 1b)." If the premise of the work is that drug-resistant TB strains are prone to mild hypermutator phenotypes, wouldn't one expect an elevated number of SNPs in DR strains? Probably need to compare within linaeg to nearest neighbors, sine diff lineages divereged from reference Rv at diff times

– Lines 128-130: "This result is in accord with the studies published in other bacteria, where a synonymous mutation impacts mRNA stability (26-29)." Since the authors do not measure mRNA stability in this manuscript, this statement is inaccurate.

– The reviewer would appreciate if gene common names were included as an individual column in Supplemental Tables (e.g. Table S3,S4, etc.) to facilitate evaluation of the data.

[Editors’ note: further revisions were suggested prior to acceptance, as described below.]

Thank you for resubmitting your work entitled "GWAS and functional studies suggest a role for altered DNA repair in the evolution of drug resistance in *Mycobacterium tuberculosis*" for further consideration by *eLife*. Your revised article has been evaluated by Bavesh Kana (Senior Editor) and a Reviewing Editor.

The editors and reviewers agree that the revised manuscript has been significantly improved by the incorporation of substantial new evidence; the authors are congratulated on the extent of the revisions, and the volume of new experimental and analytical data added. There is, however, one substantive issue remaining that must be addressed:

All three reviewers feel that the evidence is strong in support of the central claim that the identified mycobacterial MutY polymorphism confers increased frequency of resistance to antibiotics in vitro and impacts survival in vivo in models of *M. tuberculosis* infection. These are valuable insights. What is less certain, though, is whether this phenotype is due to hypermutagenesis, as the authors propose. In the absence of definitive evidence in support of this possibility, the authors are advised to temper any claims about mechanism; rather, the language should be softened to convey the conclusion that a polymorphism found in some clinical strains impairs MutY function, but the consequences of this for *M. tuberculosis* pathogenesis and/or emergence of drug resistance require further experimentation.

*Reviewer #1 (Recommendations for the authors):*

In my review of the original manuscript, I raised three concerns, namely that (i) the GWAS analysis was confusing; (ii) albeit tantalising, the proposed mechanism of enhanced/accelerated evolution of drug-resistance under antibiotic treatment was inadequately supported by the data presented; and (iii) it was difficult to determine whether the phenotypes reported resulted from genuine drug resistance or drug tolerance.

By incorporating significant new experimental and analytical data (for which the authors are congratulated), the revised version submitted here for review has substantially addressed these concerns.

My sole nagging doubt relates to the proposal that deficient MutY function results in hypermutagenesis, in turn allowing for selection of fitter (drug resistant) mutants under antibiotic treatment. The evidence presented in support of this conclusion remains inconclusive, despite the authors' contentions. For example, the analysis of C-T/C-G mutations (in Figure 2—figure supplement 6i) does not provide compelling evidence of defective 8-oxoG repair (it is notable, too, that these data are presented without any assessment of statistical significance). This is not a fatal doubt, however it does suggest that the authors should temper any claims about underlying mechanism – better to report the observed association of mutY mutations with propensity for drug resistance in the absence of overly speculative theories around mechanism.

*Reviewer #2 (Recommendations for the authors):*

The authors have submitted a revised manuscript in response to the prior reviews. In that review, the points raised were:

1) Provide additional data that the enhanced survival of the Mtb mutY KO in macrophages and guinea pigs is linked to hyper-mutability and consequent mutations that enhance fitness.

2) More statistical rigor in the number of replicates in the mutation frequency analysis.

3) Provide more data on strain relatedness.

The authors have responded positively to these critiques and have supplied more data.

1) The strain relatedness has been clarified.

2) The manuscript now includes mutation rate measurements which show enhanced mutation rates in the mutY KO.

3) On the question of whether the enhanced survival of the MutY KO can be attributed to enhanced fitness through mutagenesis, the authors have provided new data of whole genome sequencing of isolated colonies that arose from passage through macrophages or guinea pigs, in the former case in the presence or absence of antibiotics. I thank the authors for undertaking this extensive experimentation in response to the critique. In some cases, the data does seem to show a correlation between inactivation of mutY and evolution of antimicrobial resistance. This seems clearest for the gyrA mutation in the presence of cipro treatment (see Figure 6E) which is present in a high proportion of treated mutY null or R262Q strains, but not in wild type or complemented. This provides some support for the hypothesis that the MutY polymorphism can enhance the evolution of antimicrobial resistance, although one cannot derive true frequencies from this data.

The data for mutation accumulation in the absence of antibiotics is less clear and can't be as clearly linked to a phenotypic advantage in vivo. So, I think the genome sequencing adds to the story about evolution of drug resistance, but I don't think it supports the idea that the enhanced in vivo growth is due to adaptive evolution. I think this latter point should be softened as other explanations are possible, including a toxic effect of MutY DNA damage in the presence of oxo-G in vivo, as noted in the prior review.

Despite these caveats, I do think the paper discovers a MutY polymorphism and shows it is functionally important for mutagenesis, and in vivo survival, which is an important finding linking a clinical strain polymorphism to a mechanism of drug resistance and pathogenesis.

*Reviewer #3 (Recommendations for the authors):*

This reviewer thanks the authors for their revised submission. I have only one suggestion related to "Essential Revision" point 3 from the first review of this manuscript.

"(3) Related to point (2) above: the clinical *M. tuberculosis* strains carrying the identified mutY and uvrB mutations should contain additional polymorphisms as a consequence of the loss/impairment of core DNA repair function(s), however no evidence is presented in support of (or refuting) this assumption."

The primary conclusion of this paper is that loss-of-function mutations in the DNA repair genes uvrB and mutY contribute to the evolution of drug resistance in Mtb by elevating the mutation rate (Figure 2 —figure supplement 5). In support of this conclusion, the authors present data in M. smegmatis (uvrB) or Mtb (mutY) that two identified mutants (uvrB-A524V and mutY-R262Q) have elevated mutation rates in laboratory strains.

Where the authors work would benefit greatly is convincingly demonstrating this association- mutations in uvrB and mutY and elevated mutation rates- using publicly available whole-genome sequencing data from clinical Mtb strains. If the authors hypothesis is true, then it is reasonable to expect that clinical Mtb strains harboring loss-of-function mutations in uvrB and mutY would have more polymorphisms relative to strains WT for uvrB and mutY. The authors present a preliminary analysis to this effect and do not see this association (response to reviewers document). Figure 2 —figure supplement 6I supposedly shows this association, but this reviewer could not find a detailed explanation how this analysis was done, nor is it clear that the authors statement "showed increased C->T or C->G mutations" is true, particularly for C->G. The authors mention that lack of mutY should lead to elevated C->G or C->A mutations, so all SNP types should be included in this analysis to see if any hypermutator phenotype is specific to the SNP effects expected for mutY.

If the authors can analyze publicly available Mtb genome sequencing data and convincingly demonstrate an association between uvrB and mutY mutations and elevated polymorphism burden- or provide a convincing explanation as to why this association is not expected or seen- this would strongly support the primary conclusion of this manuscript.

---

## [Author Response]

Essential revisions:The reviewers were in general agreement that the work presents interesting and potentially important results; however, all reviewers felt that, in its current form, the conclusions are inadequately supported by the data. The following are therefore considered essential to address the major concerns:1) The GWAS analysis provides the foundation of the paper, yet all three reviewers expressed concerns about the presentation of – and, therefore, inferences derived from – this part of the study. More, and clearer, information is required about the strains included, their potential phylogenetic relatedness, and the distributions of the identified resistance-associated mutations.

We thank all the reviewers for their insightful comments. In the revised manuscript, we have performed the phylogenetic analysis of the strains used. A phylogenetic tree was generated using Mycobacterium canetti as an outgroup (Figure 1b). The phylogeny analysis suggests the clustering of the strains in lineage 1, 2, 3, and 4. Lineages 2, 3 and 4 are clustering together, and lineage 1 is monophyletic, as reported previously. The genome sequence data of 2773 clinical strains were downloaded from NCBI. These strains were also part of the GWAS analysis performed by Coll et al. (https://pubmed.ncbi.nlm.nih.gov/29358649/) and Manson et al. (https://pubmed.ncbi.nlm.nih.gov/28092681/). The phenotype of the strains used for the association analysis was reported in the previous studies. We have not performed other predictions. The supplementary table provides the lineage origin of each strain used in the study (Supplementary File 1 and 2). The distributions of resistance-associated mutations in different strains is shown (Figure 2—figure supplement 6a-h). As suggested, we have performed an analysis wherein we looked for the direct target mutations that harbor mutations in the DNA repair genes (Figure 2—figure supplement 6i-k). We have extensively worked on the presentation of the data to make it more discernible.

2) A key claim of this work is that the in vivo fitness advantage of the MutY mutant strain is due to hypermutagenesis and selection of a fitter strain, however the evidence supporting this conclusion is lacking.

To ascertain if the better survival of the RvDmutY, or RvDmutY::mutY-R262Q, is indeed due to the acquisition of mutations in the direct target of antibiotics, we performed WGS of the strain from the ex vivo evolution experiment (Figure 5). Genomic DNA extracted from ten independent colonies (grown in vitro), was mixed in equal proportion prior to library preparation. For the analysis, only those SNPs that were present in >20% of reads were retained. Analysis of Rv sequences grown in vitro suggested that the laboratory strain has accumulated 100 SNPs compared with the reference strain. The sequence of Rv laboratory strain was used as the reference strain for the subsequent analysis. WGS data for RvDmutY, RvDmutY::mutY, and RvDmutY::mutY-R262Q strains grown in vitro did not show the presence of the mutation in the antibiotic target genes. In a similar vein, ten independent colonies each from the 7H11-OADC plates after the final round of ex vivo selection in the presence or absence of antibiotics were selected for WGS. Data indicated that in the absence of antibiotics, no direct target mutations were identified in the ex vivo passaged strains (Figure 6a and e). In the presence of isoniazid, we found mutations in the katG (Ser315Thr or Ser315Ileu) in the Rv, RvDmutY but not in RvDmutY::mutY and RvDmutY::mutY-R262Q (Figure 6b and e). These findings are in congruence with the ex vivo evolution CFU analysis, wherein we did not observe a significant increase in the survival of RvDmutY and RvDmutY::mutY-R262Q in the presence of isoniazid (Figure 5). In the presence of ciprofloxacin and rifampicin, direct target mutations were identified in the gyrA and rpoB (Figure 6c-e). Asp94Glu/Asp94Gly mutations were identified in gyrA, and, His445Tyr/Ser450Leu mutations were identified in rpoB of RvDmutY and RvDmutY::mutY-R262Q, respectively. No direct target mutations were identified in the Rv and RvDmutY::mutY, suggesting that the perturbed DNA repair aids in acquiring the drug resistance-conferring mutations in Mtb (Figure 6c-e and Supplementary File 8).

To determine if the better survival of the RvDmutY, or RvDmutY::mutY-R262Q, in the guinea pig infection experiment (Figure 8) is due to the accumulation of mutations in the host, we performed WGS of the strain isolated from guinea pig lungs. Analysis revealed specific genes such as cobQ1, smc, espI, and valS were mutated only in RvDmutY and RvDmutY::mutYR262Q but not in Rv and RvDmutY::mutY. Besides, tcrA and gatA were mutated only in RvDmutY, whereas rv0746 were mutated exclusively in the RvDmutY::mutY (Figure 8—figure supplement 2). However, we did not observe any direct target mutations; this may be because guinea pigs were not subjected to antibiotic treatment. Data suggests that the continued longterm selection pressure is necessary for bacilli to acquire mutations.

3) Related to point (2) above: the clinical M. tuberculosis strains carrying the identified mutY and uvrB mutations should contain additional polymorphisms as a consequence of the loss/impairment of core DNA repair function(s), however no evidence is presented in support of (or refuting) this assumption.

We analyzed the genome of the clinical strains that possess DNA repair gene mutations to determine the additional polymorphisms. The number of SNPs in the strains harboring DNA repair mutation and the drug-susceptible strains appears to be similar. The marginal difference, if any were not statistically significant.

**Author response image 1. sa2fig1:** 

4) There was some uncertainty among the reviewers whether the phenotypes reported are genuine drug-resistance, or rather a form of (heritable) tolerance. This uncertainty, which might be ascribed to the use throughout the manuscript of mutation frequency assessments, not mutation rates, must be resolved. For example, through the use of fluctuation assays; by determining resistance levels quantitatively (i.e., in MIC assessments); and/or by establishing the breadth of antimycobacterial agents (cidal and static drugs with different targets/mechanisms of action) to which reduced susceptibility is conferred by the identified DNA repair polymorphisms.

As suggested, we determined the mutation rate in the presence of isoniazid, rifampicin, and ciprofloxacin (Figure 3g-j). The fold increase in the mutation rate relative to Rv for RvDmutY, RvDmutY:mutY, and RvDmutY::mutY-R262Q was 2.90, 0.76, and 3.0 in the presence of isoniazid and 5.62, 1.13, and 5.10 or 9.14, 1.57, and 8.71 in the presence of rifampicin and ciprofloxacin respectively.

In addition, we determined the effect of different drugs on the survival of RvDmutY or RvDmutY::mutY-R262Q by performing killing kinetics in the presence and absence of isoniazid, rifampicin, ciprofloxacin, and ethambutol (Figure 4a). In the absence of antibiotics, the growth kinetics of Rv, RvDmutY, RvDmutY:mutY, and RvDmutY::mutY-R262Q were similar (Figure 4b). In the presence of isoniazid, ~2 log-fold decreases in bacterial survival was observed on day 3 in Rv and RvDmutY:mutY; however, in RvDmutY and RvDmutY::mutY-R262Q, the difference was limited to ~1.5 log-fold (Figure 4c). A similar trend was apparent on days 6 and 9, suggesting a ~5-fold increase in the survival of RvDmutY and RvDmutY::mutY-R262Q compared with Rv and RvDmutY:mutY (Figure 4c). Interestingly, in the presence of ethambutol, we did not observe any significant difference (Figure 4d). In the presence of rifampicin and ciprofloxacin, we observed a ~10-fold increase in the survival of RvDmutY and RvDmutY::mutY-R262Q compared with Rv and RvDmutY:mutY (Figure 4e-f). Thus results suggest that the absence of mutY or the presence of mutY variant aids in subverting the antibiotic stress.

Reviewer #1 (Recommendations for the authors):This manuscript presents intriguing data suggesting a role for defective DNA repair pathways in the development of drug resistance under treatment. However, there are a few key issues/questions which must be addressed.(i) The GWAS analysis is confusing and/or unclear as described:a. The authors report (L85-7) the use of known drug resistance phenotypes (where available) or inferences of drug-resistance from genotypic data to enhance the potential to identify other mutations that might be implicated in enabling the DR mutations, yet their list of known DR mutations (Table 1) seems predominantly to comprise rare or unusual mutations, not those commonly associated with clinical DR-TB.

We have modified the manuscript extensively to make it more discernible. In the revised manuscript, we have performed the phylogenetic analysis of the strains used. A phylogenetic tree was generated using Mycobacterium canetti as an outgroup (Figure 1b). The phylogeny analysis suggests the clustering of the strains in lineage 1, 2,3, and 4. Lineages 2,3 and 4 are clustering together, and lineage 1 is monophyletic, as reported previously. The genome sequence data of 2773 clinical strains were downloaded from NCBI. These strains were also part of the GWAS analysis performed by Coll et al. (https://pubmed.ncbi.nlm.nih.gov/29358649/) and Manson et al.

(https://pubmed.ncbi.nlm.nih.gov/28092681/). The phenotype of the strains used for the association analysis was reported in the previous studies. We have not performed other predictions. The supplementary table provides the lineage origin of each strain used in the study (Supplementary File 1 and 2). The distributions of resistance-associated mutations in different strains is shown (Figure 2—figure supplement 6 a-h). As suggested, we have performed an analysis wherein we looked for the direct target mutations that harbor mutations in the DNA repair genes (Figure 2—figure supplement 6 i-k).

We identified mostly the rare mutations due to the following reasons;

1. We looked for the mutations that were present only in the multidrug resistant strains as compared to the susceptible strains for association mapping. This strategy exclusively gave most variants associated with multidrug resistant phenotype.

2. We have used Mixed Linear Model (MLM) for association analysis. MLM removes all the population-specific SNPs based on PCA and kinship corrections. The false discovery rate (FDR) adjusted p-values in the GAPIT software are stringent as it corrects the effects of each marker based on the population structure (Q) as well as kinship (K) values. Therefore the probability of identifying the false-positive SNP is very low. We combined it with the Bonferroni corrections to identify markers associated with the drug resistant phenotype.

b. In the same lines (85-7), the authors state that strains were separated into 5 drug susceptibility categories, the last of which was "pre-XDR", yet later sentences (e.g., L95, 104, 128, etc.) refer to "MDR/XDR" strains. Are these XDR or pre-XDR?

Strains used for association analysis were drug susceptible, MDR, Poly-DR, pre-XDR, and XDR. MDR/XDR refers to all the categories put together.

c. The distributions of the identified resistance-associated mutations across the different lineages need to be explained more clearly; this is necessary to strengthen claims of specific selection of the mutations under antibiotic treatment.

As suggested, we have reported the lineage of identified resistant-associated mutation (Figure 2—figure supplement 6i).

(ii) A central claim of the paper is that enhanced fitness, as a consequence of elevated mutagenesis, contributes to the prevalence of the uvrB and mutY mutations among the DR Mtb strains. While plausible, this is not explored in the manuscript: instead, the experimental work is limited to assessments of competitive survival in various models, with/without antibiotic selection, or to mutant frequency analyses. There is no direct evidence of increased mutational load in the mutY and uvrB strains. To strengthen the credibility of this central claim, WGS data of "successful"/DR mutants would be very useful.

To ascertain if the better survival of the RvDmutY, or RvDmutY::mutY-R262Q, is indeed due to the acquisition of mutations in the direct target of antibiotics, we performed WGS of the strain from the ex vivo evolution experiment (Figure 5). Genomic DNA extracted from ten independent colonies (grown in vitro) was mixed in equal proportion prior to library preparation. Only those SNPs present in >20% of reads were retained for the analysis. Analysis of Rv sequences grown in vitro suggested that the laboratory strain has accumulated 100 SNPs compared with the reference strain. The sequence of Rv laboratory strain was used as the reference strain for the subsequent analysis. WGS data for RvDmutY, RvDmutY::mutY, and RvDmutY::mutY-R262Q strains grown in vitro did not show the presence of a mutation in the antibiotic target genes. In a similar vein, ten independent colonies, each from the 7H11-OADC plates, after the final round of ex vivo selection in the presence or absence of antibiotics, were selected for WGS. Data indicated that in the absence of antibiotics, no direct target mutations were identified in the ex vivo passaged strains (Figure 6a and e). In the presence of isoniazid, we found mutations in the katG (Ser315Thr or Ser315Ileu) in the Rv, RvDmutY but not in RvDmutY:mutY and RvDmutY::mutY-R262Q (Figure 6b and e). These findings are in congruence with the ex vivo evolution CFU analysis, wherein we did not observe a significant increase in the survival of RvDmutY and RvDmutY::mutY-R262Q in the presence of isoniazid (Figure 5). In the presence of ciprofloxacin and rifampicin, direct target mutations were identified in the gyrA and rpoB (Figure 6c-e).

Asp94Glu/Asp94Gly mutations were identified in gyrA, and, His445Tyr/Ser450Leu mutations were identified in rpoB of RvDmutY and RvDmutY::mutY-R262Q, respectively. No direct target mutations were identified in the Rv and RvDmutY::mutY, suggesting that the perturbed DNA repair aids in acquiring the drug resistance-conferring mutations in Mtb (Figure 6c-e and Supplementary File 8).

To determine if the better survival of the RvDmutY, or RvDmutY::mutY-R262Q, in the guinea pig infection experiment (Figure 8) is due to the accumulation of mutations in the host, we performed WGS of the strain isolated from guinea pig lungs. Analysis revealed specific genes such as cobQ1, smc, espI, and valS were mutated only in RvDmutY and RvDmutY::mutYR262Q but not in Rv and RvDmutY::mutY. Besides, tcrA and gatA were mutated only in RvDmutY, whereas rv0746 were mutated exclusively in the RvDmutY:mutY (Figure 8—figure supplement 2). However, we did not observe any direct target mutations; this may be because guinea pigs were not subjected to antibiotic treatment. Data suggests that the continued longterm selection pressure is necessary for bacilli to acquire mutations.

(iii) Continuing from the above, it appears that these mutations confer heritable tolerance, rather than resistance. This does not undermine the value of this work, but it does suggest additional lines of experimental inquiry, for example:a. mutation rate (not frequency) assessments;

In the revised manuscript, we have performed the mutation rate analysis in the presence of different drugs (Figure 3g-j).

b. use of higher drug concentrations (e.g., rifampicin at 200 ug/ml in M. smegmatis);

We thank the reviewer for the comment. We initially used 100 µg/ml conc. of rifampicin. However, we did not get any colonies on the plates. Thus we chose 50 µg/ml rifampicin concentration in Msm for performing experiments.

c. MIC assays to indicate precisely (quantitatively) the impact of the mutations on drug susceptibility;

We thank the reviewer for the comment. In the revised manuscript, we have performed killing kinetics in the absence and presence of rifampicin, isoniazid, ethambutol and ciprofloxacin.

d. expansion of the drug panel to include compounds with different mechanisms of action, cidal as well as static; that is, how broadly tolerant are these strains as a consequence of the individual mutations?

To evaluate the effect of different drugs on the survival of RvDmutY or RvDmutY::mutYR262Q, we performed killing kinetics in the presence and absence of isoniazid, rifampicin, ciprofloxacin, and ethambutol (Figure 4a). In the absence of antibiotics, the growth kinetics of Rv, RvDmutY, RvDmutY::mutY, and RvDmutY::mutY-R262Q were similar (Figure 4b). In the presence of isoniazid, ~2 log-fold decreases in bacterial survival was observed on day 3 in Rv and RvDmutY:mutY; however, in RvDmutY and RvDmutY::mutY-R262Q, the difference was limited to ~1.5 log-fold (Figure 4c). A similar trend was apparent on days 6 and 9, suggesting a ~5-fold increase in the survival of RvDmutY and RvDmutY::mutY-R262Q compared with Rv and RvDmutY::mutY (Figure 4c). Interestingly, in the presence of ethambutol, we did not observe any significant difference (Figure 4d). In the presence of rifampicin and ciprofloxacin, we observed a ~10-fold increase in the survival of RvDmutY and RvDmutY::mutY-R262Q compared with Rv and RvDmutY:mutY (Figure 4e-f). Thus results suggest that the absence of mutY or the presence of mutY variant aids in subverting the antibiotic stress.

(iv) Continuing from the above, increased mutagenesis is generally deleterious, especially where enabled by the loss of core DNA repair functions; it would be useful to know, therefore, whether the mutY and uvrB mutations – both genes encoding proteins involved in excision repair pathways – incur any fitness costs. This could be assessed via DNA damage tolerance/survival assays. Also, are both mutations tolerated in a single strain (is there any evidence of this in the GWAS data?)? That is, do dual mutY uvrB alleles enhance drug survival, or reduce it, compared to the single mutants alone?

We thank the reviewer for the insightful comment. We have evaluated the survival of Rv, RvDmutY, RvDmutY::mutY, and RvDmutY::mutY-R262Q the presence of oxidative and nitrosative stress. We did not observe any significant decrease in the survival of the mutants compared with the wild type (data not shown).

Moreover, we have not observed both mutations on a single strain in our GWAS data. While it is possible that the presence of both mutations may be beneficial for the organism, we have not made any attempts to generate double mutant.

(v) Why is the wild-type parental strain (Rv) selectable on kanamycin-containing media? Presumably this is because it contains the complementation vector?

The wild-type parental strain (Rv) has a kanamycin resistance-containing complementation vector.

Reviewer #3 (Recommendations for the authors):Text improvements:– It is curious that this GWAS analysis did not identify some of the most common acquired drug-resistance conferring mutations. For example, as the authors note, this analysis identifies rpoB p.Leu452Pro (note that in the main text this is mislabeled as p.Leu452Val) and p.Val496Met, but misses the far more common Ser450L and Asp435Val alleles. A discussion as to why these (and other) known drug resistance mutations did not meet the cutoffs would be welcomed.– Lines 49-50: "Although Mtb has a lower mutation rate…" Lower than what?

We have removed the sentence from the revised manuscript.

– Lines 56-59: "Despite well-known mechanisms of drug resistance, in 10-40% of the clinical isolates of Mtb, drug resistance cannot be determined by the mutations in the direct targets of antibiotics, implying the presence of hitherto unknown mechanisms that foster the development of resistance in Mtb (8)." It would be worth updating this statement based on the most recent work from the CRYPTIC consortium, for example.

As suggested, we have incorporated the changes in the revised manuscript.

– Lines 89-91: "The total number of SNPs observed for susceptible, mono-DR, MDR, or pre-XDR strains were comparable, suggesting no genetic drift during the evolution of antibiotic resistance (Figure 1b)." If the premise of the work is that drug-resistant TB strains are prone to mild hypermutator phenotypes, wouldn't one expect an elevated number of SNPs in DR strains? Probably need to compare within linaeg to nearest neighbors, sine diff lineages divereged from reference Rv at diff times

We analyzed the genome of the clinical strains that possess DNA repair gene mutations to determine the additional polymorphisms. The number of SNPs in the strains harboring DNA repair mutation and the drug-susceptible strains appears to be higher. We have also looked for the CàT and CàG mutations in the same strains. CàT mutations are higher in the strains harboring mutY variant compared with the susceptible strains (Figure 2—figure supplement 6l). However, we could not perform statistical analysis as the number of strains harbor mutY variant is limited to 8. Thus data suggest that empirically ^mutY^
_phenotype_
^sensitive^ the strains harboring mutY variant show higher SNPs elsewhere and CàT mutations. However, we are not stating these conclusions very strongly in the manuscript as the data is not statistically significant.

– Lines 128-130: "This result is in accord with the studies published in other bacteria, where a synonymous mutation impacts mRNA stability (26-29)." Since the authors do not measure mRNA stability in this manuscript, this statement is inaccurate.

As suggested, we have removed the line from the revised manuscript.

– The reviewer would appreciate if gene common names were included as an individual column in Supplemental Tables (e.g. Table S3,S4, etc.) to facilitate evaluation of the data.

As suggested, we have incorporated the changes in the revised manuscript. These tables are labelled as Supplementary File 3 and 4.

[Editors' note: further revisions were suggested prior to acceptance, as described below.]

The editors and reviewers agree that the revised manuscript has been significantly improved by the incorporation of substantial new evidence; the authors are congratulated on the extent of the revisions, and the volume of new experimental and analytical data added. There is, however, one substantive issue remaining that must be addressed:All three reviewers feel that the evidence is strong in support of the central claim that the identified mycobacterial MutY polymorphism confers increased frequency of resistance to antibiotics in vitro and impacts survival in vivo in models of M. tuberculosis infection. These are valuable insights. What is less certain, though, is whether this phenotype is due to hypermutagenesis, as the authors propose. In the absence of definitive evidence in support of this possibility, the authors are advised to temper any claims about mechanism; rather, the language should be softened to convey the conclusion that a polymorphism found in some clinical strains impairs MutY function, but the consequences of this for M. tuberculosis pathogenesis and/or emergence of drug resistance require further experimentation.

As suggested, we have worked on the language and temper claims about the mechanism. The changes are incorporated and shown with the revised manuscript's track change.

Reviewer #1 (Recommendations for the authors):In my review of the original manuscript, I raised three major concerns, namely that (i) the GWAS analysis was confusing; (ii) albeit tantalising, the proposed mechanism of enhanced/accelerated evolution of drug-resistance under antibiotic treatment was inadequately supported by the data presented; and (iii) it was difficult to determine whether the phenotypes reported resulted from genuine drug resistance or drug tolerance.By incorporating significant new experimental and analytical data (for which the authors are congratulated), the revised version submitted here for review has substantially addressed these concerns.My sole nagging doubt relates to the proposal that deficient MutY function results in hypermutagenesis, in turn allowing for selection of fitter (drug resistant) mutants under antibiotic treatment. The evidence presented in support of this conclusion remains inconclusive, despite the authors' contentions. For example, the analysis of C-T/C-G mutations (in Figure 2—figure supplement 6i) does not provide compelling evidence of defective 8-oxoG repair (it is notable, too, that these data are presented without any assessment of statistical significance). This is not a fatal doubt, however it does suggest that the authors should temper any claims about underlying mechanism – better to report the observed association of mutY mutations with propensity for drug resistance in the absence of overly speculative theories around mechanism.

As suggested, we have worked on the language and temper claims about mechanism. The changes are incorporated and shown with the track change (highlighted in blue) in the revised manuscript.

Reviewer #2 (Recommendations for the authors):The authors have submitted a revised manuscript in response to the prior reviews. In that review, the major points raised were:1) Provide additional data that the enhanced survival of the Mtb mutY KO in macrophages and guinea pigs is linked to hyper-mutability and consequent mutations that enhance fitness.2) More statistical rigor in the number of replicates in the mutation frequency analysis.3) Provide more data on strain relatedness.The authors have responded positively to these critiques and have supplied more data.1) The strain relatedness has been clarified.2) The manuscript now includes mutation rate measurements which show enhanced mutation rates in the mutY KO.3) On the question of whether the enhanced survival of the MutY KO can be attributed to enhanced fitness through mutagenesis, the authors have provided new data of whole genome sequencing of isolated colonies that arose from passage through macrophages or guinea pigs, in the former case in the presence or absence of antibiotics. I thank the authors for undertaking this extensive experimentation in response to the critique. In some cases, the data does seem to show a correlation between inactivation of mutY and evolution of antimicrobial resistance. This seems clearest for the gyrA mutation in the presence of cipro treatment (see Figure 6E) which is present in a high proportion of treated mutY null or R262Q strains, but not in wild type or complemented. This provides some support for the hypothesis that the MutY polymorphism can enhance the evolution of antimicrobial resistance, although one cannot derive true frequencies from this data.The data for mutation accumulation in the absence of antibiotics is less clear and can't be as clearly linked to a phenotypic advantage in vivo. So, I think the genome sequencing adds to the story about evolution of drug resistance, but I don't think it supports the idea that the enhanced in vivo growth is due to adaptive evolution. I think this latter point should be softened as other explanations are possible, including a toxic effect of MutY DNA damage in the presence of oxo-G in vivo, as noted in the prior review.Despite these caveats, I do think the paper discovers a MutY polymorphism and shows it is functionally important for mutagenesis, and in vivo survival, which is an important finding linking a clinical strain polymorphism to a mechanism of drug resistance and pathogenesis.

As suggested we have softened the explanation regarding the enhanced in vivo growth of different strains (line 309-312).

Reviewer #3 (Recommendations for the authors):This reviewer thanks the authors for their revised submission. I have only one major suggestion related to "Essential Revision" point 3 from the first review of this manuscript."(3) Related to point (2) above: the clinical M. tuberculosis strains carrying the identified mutY and uvrB mutations should contain additional polymorphisms as a consequence of the loss/impairment of core DNA repair function(s), however no evidence is presented in support of (or refuting) this assumption."The primary conclusion of this paper is that loss-of-function mutations in the DNA repair genes uvrB and mutY contribute to the evolution of drug resistance in Mtb by elevating the mutation rate (Figure 2 —figure supplement 5). In support of this conclusion, the authors present data in M. smegmatis (uvrB) or Mtb (mutY) that two identified mutants (uvrB-A524V and mutY-R262Q) have elevated mutation rates in laboratory strains.Where the authors work would benefit greatly is convincingly demonstrating this association- mutations in uvrB and mutY and elevated mutation rates- using publicly available whole-genome sequencing data from clinical Mtb strains. If the authors hypothesis is true, then it is reasonable to expect that clinical Mtb strains harboring loss-of-function mutations in uvrB and mutY would have more polymorphisms relative to strains WT for uvrB and mutY. The authors present a preliminary analysis to this effect and do not see this association (response to reviewers document). Figure 2 —figure supplement 6I supposedly shows this association, but this reviewer could not find a detailed explanation how this analysis was done, nor is it clear that the authors statement "showed increased C->T or C->G mutations" is true, particularly for C->G. The authors mention that lack of mutY should lead to elevated C->G or C->A mutations, so all SNP types should be included in this analysis to see if any hypermutator phenotype is specific to the SNP effects expected for mutY.If the authors can analyze publicly available Mtb genome sequencing data and convincingly demonstrate an association between uvrB and mutY mutations and elevated polymorphism burden- or provide a convincing explanation as to why this association is not expected or seen- this would strongly support the primary conclusion of this manuscript.

The analysis presented in response to the reviewers' document and Figure 2—figure supplement 6I was performed using available *Mtb* genome sequencing data of clinical strains. To analyze the mutation spectrum, we used the same clinical strains which are used in the present study. As suggested, we have included all the SNP types in the analysis. We have identified a higher trend towards CA, AG, and CT mutations in the strains harboring *mutY* polymorphism compared with closely related drug-susceptible strains. However, we could not perform the statistical analysis because the number of strains harboring *mutY* mutation was limited. We have added this analysis in Figure 2—figure supplement 6l and mentioned it in the figure legend and discussion. Analysis of other publicly available databases for *uvrB* or *mutY* mutations and possible implications on elevated polymorphism would be attempted subsequently.